# Sediment export in marly badland catchments modulated by frost-cracking intensity, Draix-Bléone Critical Zone Observatory, SE France.

Coline Ariagno[1], Caroline Le Bouteiller[1], Peter van der Beek[2] and Sébastien Klotz[1]

[1]Univ. Grenoble Alpes, INRAE, UR ETNA, Grenoble, France

[2]Institut für Geowissenschaften, Universität Potsdam, Germany

*Correspondence to*: Coline Ariagno (coline.ariagno@inrae.fr)

**Abstract.** At the interface between the lithosphere and the atmosphere, the critical zone records the complex interactions between erosion, climate, geologic substrate and life, and can be directly monitored. Long data records (30 consecutive years for sediment yields) collected in the sparsely vegetated, steep and small marly badland catchments of the Draix-Bléone Critical Zone Observatory (CZO), SE France, allow analysing potential climatic controls on regolith dynamics and sediment export. Although widely accepted as a first-order control, rainfall variability does not fully explain the observed inter-annual variability in sediment export. Previous studies in this area have suggested that frost-weathering processes could drive regolith production and potentially modulate the observed pattern of sediment export. Here, we define sediment-export anomalies as the residuals from a predictive model with annual rainfall intensity above a threshold as the control. We then use continuous soil-temperature data, recorded at different locations over multiple years, to highlight the role of different frost weathering processes (i.e., ice segregation versus volumetric expansion) in regolith production. Several proxies for different frost-weathering processes have been calculated from these data and compared to the sediment-export anomalies, with careful consideration of field data quality. Our results suggest that frost-cracking intensity (linked to ice segregation) can explain about half (47-64%) of the sediment-export anomalies. In contrast, the number of freeze-thaw cycles (linked to volumetric expansion) has only a minor impact on catchment sediment response. The time spent below 0 °C also correlates well with the sediment-export anomalies and requires fewer field data to be calculated than the frost-cracking intensity. Thus, frost-weathering processes modulate sediment export by controlling regolith production in these catchments and should be taken into account when building predictive models of sediment export from these badlands under a changing climate.

## 1. Introduction

Landscape erosion and its associated hazards, such as torrential floods, rockfalls, etc., are some of the visible consequences of the complex interaction between the Critical Zone and climate (e.g., Anderson et al., 2012). Regolith production is the result of a variety of processes, many of which are influenced by climate, and constitutes a critical first step in the source-to-sink

sediment pathway (e.g. Dixon et al., 2009; Riebe et al., 2017). However, the impact of climate (change) on regolith production and ensuing landscape erosion remains difficult to quantify. In a context of rapid global climate change, how will sediment production be affected (e.g., Nearing et al., 2004; Hirschberg et al., 2021; Nadal-Romero et al., 2021)? How will dominant erosion processes evolve according to lithologies and climatic variations? How will vegetation evolve locally and will it amplify or reduce surface erosion? These questions, among others, motivate the study of soil weathering mechanisms and their sensitivity to climate (e.g., Nadal-Romero et al., 2018).

In badland areas, the interaction between erosion and climate is enhanced because of the absence of vegetation and the easily erodible lithologies. Following the general definition, "badlands" refers to "deeply dissected erosional landscapes, formed in soft-rock terrain, commonly but not exclusively in semi-arid regions and with sparse vegetation, that have a high drainage density of rill and gully systems and are dominated by overland flow" (Harvey, 2004). The widespread badland landscapes known as "Terres Noires" in the South-Eastern French Alps have been extensively studied because of their susceptibility to erosion, leading to high sediment export (e.g., Antoine et al., 1995). Over the last 35 years, several small catchments in these marly badlands have been monitored in the framework of the Draix–Bléone Critical-Zone Observatory (CZO), leading to a quantification of weathering and erosion through repeated measurements of sediment yield at the event scale (Mathys et al., 2003). Because of the ample availability of sediment and the efficient network connectivity (Jantzi et al., 2017), floods in these catchments can transport a very large quantity of sediment (Delannoy and Rovéra, 1996). As an example, during one flood event on 17/06/2014, 6390 tonnes/km² were exported and the suspended sediment concentration reached 440 g/l. Such sediment-laden floods can potentially cause significant damage to downstream infrastructure. Landscape changes are easily and rapidly observable in the Draix–Bléone catchments, but improved identification and understanding of the weathering processes in these marls are required to more accurately predict exported sediment volumes.

Several studies have addressed the characteristics and dynamics of regolith development in the marly badlands of the Draix-Bléone CZO and similar sites. The impact of water content, hydraulic conductivity and infiltration capacity of marls in the Draix-Bléone CZO on runoff generation and erosion was studied by Esteves et al. (2005), Mathys et al., 2005 and Mallet et al., 2018. Rovéra and Robert (2005) first investigated periglacial erosion processes in the Draix-Bléone CZO; they noted the marls' sensitivity to frost weathering, in particular to freeze-thaw cycles, and the resulting faster ablation on north-facing compared to south-facing slopes. Working in the Central and Eastern Spanish Pyrenees, Regüés et al. (1995) and Nadal-Romero et al. (2007) confirmed the important role of slope aspect in controlling the weathering of marls. By studying bulk density, surface mechanical resistance and moisture content, they highlighted a clear temporal and spatial variability in the regolith and inferred that weathering in these catchments was mainly dependent on the number of freeze-thaw cycles occurring during the year. Based on a two-year time series of twelve high-resolution digital elevation models from a 0.13 ha catchment in the Draix-Bléone CZO, Bechet et al (2016) showed that erosion processes follow a seasonal cycle, with accumulation of loose regolith on slopes during winter followed by its transport from the slopes to the main gullies during summer. These authors inferred a yearly cycle between transport-limited conditions in spring to supply-limited conditions in autumn. However, Jantzi et al. (2017) used sediment-budget calculations from the larger Moulin and Laval catchments of the Draix-

Bléone CZO (see below) to infer that sediment transfer is not immediate; they calculated a 3-year residence time for sediments in these catchments.

Overall, existing observations from the Draix-Bléone CZO and similar sites (e.g., Regüés et al., 1995; Nadal-Romero et al., 2007) lead to consider frost-weathering as a potentially important process controlling regolith production in marly Alpine badlands. Several studies have explored this process in different geological contexts, employing theoretical, experimental and observational approaches. The two main frost-weathering processes considered in the literature are volumetric expansion of ice and ice segregation (e.g., Matsuoka, 2008). Volumetric expansion of ice can occur repeatedly during freeze-thaw cycles,

whereas ice segregation occurs when liquid water migrates towards a locus of ice-lense growth. In cold and high Alpine environments, it appears that ice segregation controls rock weathering by widening rock fractures or "frost cracking" (Anderson, 1998; Matsuoka and Murton, 2008). The growth of ice lenses required for frost cracking has been argued to depend primarily on absolute temperature, water availability and temperature gradient (Hallet et al., 1991). Hales and Roering (2007) developed a numerical model to estimate soil temperature at different depths and defined a frost-cracking intensity indicator

to quantify ice-driven mechanical erosion. In their model, frost cracking occurs when rocks are in the "frost-cracking window" (between -3 and -8 °C) and liquid water is available because either the surface or rocks at depth are above freezing point; the frost-cracking intensity depends on the temperature gradient in the frost-cracking window. Subsequently, Anderson et al. (2013) modelled rock damage by frost cracking using the above model, adding the distance that water must travel to reach the locus of potential frost cracking, and observed the impact on regolith production and hillslope evolution. Frost-cracking has

thus been identified as the major control on rock weathering in high Alpine environments (Hales and Roering, 2007; Delunel et al., 2010; Bennett et al., 2013; Draebing and Krautblatter, 2019).

The above models have, however, not yet been applied in more temperate/humid climates and in soft lithologies. Two previous studies invoking frost weathering in these contexts (Rovéra and Robert, 2005; Nadal-Romero et al., 2007) only addressed the number of freeze-thaw cycles, and thus implicitly frost weathering by volumetric expansion. Additionally, the link between

regolith production and sediment yield is known but remains difficult to quantify. Rengers et al. (2020) recently bridged this gap by studying plot-scale (22 m²) sediment accumulation and debris-flow channel filling patterns using repeat topographic surveys. They found a strong correlation between frost-cracking intensity and sediment production in a steep Alpine bedrock setting.

In this study, we develop a similar approach at the catchment scale (0.1-1 km²) in marly badlands, taking advantage of the

exceptional long-term dataset available for the Draix-Bléone Observatory. At the catchment scale, sediment export is primarily driven by rainfall, particularly during high-intensity events (Mathys et al., 2003), but we hypothesize that frost-weathering processes can modulate sediment yield, even at this scale, by controlling regolith production on hillslopes. Coupled to this first hypothesis, this study aims to highlight and quantify the main frost-weathering process in a setting of humid climate and soft lithology, by using high-resolution soil-temperature measurements (every 10 min.) from different locations. We compare

calculated temperature indicators, including the number of freeze-thaw cycles, the time spent below 0 °C, the mean negative temperature and the frost-cracking intensity, during a winter season to the sediment-export anomaly (i.e., the residual of

sediment yield that cannot be explained by rainfall variability) in the following year (e.g., for the 2007-2008 winter season, we used the sediment export for the year 2008). The goal of our study is threefold: (1) to confirm the seasonal variability of regolith supply by analysing monthly total sediment yield; (2) to quantify the role of frost-weathering processes in yearly sediment production; and (3) to identify the relevant weathering processes by statistical analysis of different proxy indicators.

## 2. Study site

The Draix-Bléone CZO is part of the French network for the study of the critical zone OZCAR (Gaillardet et al., 2018) and is specifically dedicated to the study of erosion and sediment transport in a mountainous region. Hydrological and sedimentary fluxes have been monitored on several catchments in this CZO since 1983 (Draix-Bleone Observatory, 2015). The Draix-Bléone CZO is situated in the Alpine foothills, 12 km northeast of the town of Digne in South-East France (Fig. 1). The Draix catchments are drained by the Bouinenc stream, a tributary of the Bléone, which is itself one of the main tributaries of the Durance River. The geology is characterised by Mesozoic sediments that were folded and faulted during the Alpine orogeny (Lemoine et al., 1986; Lickorish and Ford, 1998). The Bouinenc catchment has a mountainous and Mediterranean climate. Due to its relatively high elevation (>800 m above sea level), the mountainous influence is responsible for cold winters with frequent frost. The Mediterranean influence leads to dry summers interspersed by intense thunderstorms. Annual rainfall is about 900 mm/yr (varying between ∼600 and ∼1300 mm/yr interannually) in Draix. The rainfall regime varies across seasons, with high-intensity rainfall events during spring/summer and lower-intensity but longer rainfall events in autumn. Only the main streams (Laval and Moulin) are permanent, although the Moulin shows very low discharge in summer; all tributary gullies are ephemeral. Snow fall occurs almost every year but in small amounts (<10 cm) and it melts quickly. The mean annual temperature is 10.3 °C, with an annual variability between mean daily temperatures of approximatively 0.5 °C in winter and 20 °C in summer.

This study focuses on two instrumented catchments of the Draix-Bléone CZO: the Laval and the Moulin catchments, which have drainage areas of 0.86 and 0.10 km², respectively. Elevations range from 850 to 1250 m a.s.l. for the Laval and from 850 to 925 m a.s.l. for the Moulin catchment. Vegetation cover is 46% in the Moulin and 32% in the Laval catchment (Carriere et al., 2020). The catchments are underlain by thick Middle-Jurassic black marl series locally known as "Terres Noires". Subtle changes in composition of the marls, with a limestone fraction varying according to stratigraphic level, have been observed (Brochot, 1997) but our study site is not affected by these variations. The dominant bedding dip is to the east in the Draix area. These black marls are susceptible to strong erosion and develop steep badland slopes (mean hillslope angles of 0.58 for the Laval and 0.40 for the Moulin catchment), with high drainage density and deeply incised gullies characterizing the catchment morphologies (Fig. 1C, D). Sediment transport occurs through gravitational processes on hillslopes, minor landslides (<1 m$^3$) and debris-flows in the upper network, and fluvial transport as bedload and suspended load in the main network. The Draix catchments record some of the highest observed specific sediment yields worldwide: average annual sediment yields are around 12,000 and 570 tonnes, equating to specific sediment yields of around 14,000 and 5,700 tonnes/km$^2$/y, for the Laval and

Moulin catchments, respectively. This sediment budget results from 22 floods per year on average (for the Laval), ranging between 13 and 45 floods / yr and associated with very heterogeneous sediment yields from 0 up to around 6500 t/km² per event (Smetanová et al., 2018). It has been shown that erosion is strongly focused in the unvegetated parts of these catchments; the specific sediment yield of the adjacent vegetated Brusquet catchment is two orders of magnitude smaller than that of the Laval catchment (Carrière et al., 2020). Considering only the unvegetated parts of the catchments as contributing to the sediment yield and the measured sediment density of 1700 kg/m$^3$, the average erosion rate is around 8 mm/yr (Mathys, 2006).

## 3. Methods

### 3.1 Data acquisition

We use three types of datasets: rainfall, sediment yield and soil temperature. Monthly and annual rainfall values are obtained by summing the detailed event records measured with a tipping-bucket rain gauge located at the outlet of the Laval catchment (Fig. 1C). Sediment yield is measured at the outlets of both the Laval and Moulin catchments, where hydro-sedimentary stations have been set up to monitor water discharge and both suspended load and bedload. Suspended sediment concentration is measured with automatic samplers and turbidimeters and suspended sediment flux is computed as the product of discharge and concentration, cumulated over a flood to obtain an event-scale yield. Bedload volumes are measured after each flood by topographic surveys of a sediment trap located immediately upstream of the station. Bedload volume is then converted into mass using a density of 1700 kg/m$^3$, constrained by measurements in the sediment trap (Mathys, 2006). The raw data we use is therefore a series of event-scale sediment yield. An analysis of inter-event sediment export, assuming an inter-event concentration of 0.1g/L (respectively 1g/L) shows that flood export represents more than 99% (respectively 98%) of the total annual sediment export. Thus, sediment export (both as suspended load and as bedload) is considered negligible during low flow and we define the total sediment export as the monthly or yearly sum of the suspended load and bedload contributions during floods. For a few flood events, the suspended-load data is missing. In such cases, we reconstructed the event-scale suspended sediment yield based on the average proportions of suspended load and bedload, computed from multiple complete years of total load records. The proportions of suspended load and bedload in the Laval catchment are 74% / 26% for summer floods (May to September) and 57% / 43% for winter floods (October to April).

Soil temperature has been recorded continuously between August 2005 and December 2020, using several PT100 soil-temperature probes located on opposite slopes (867 m elevation) in an inner meander of the Moulin Creek (Fig. 1C, D). The acquisition frequency was 10 minutes. We use the data from probes located in bare black marls at uphill and downhill locations on north- and south-facing slopes, respectively (i.e., four sites in total; Fig. 1D), in order to explore variations in soil temperature due to differences in exposure (Rovéra and Robert, 2005). At each site, four probes are available to measure soil temperature at depths of 1, 6, 12 and 24 cm, respectively (Suppl. Fig. 1), spanning the range of depths that has been reported for the weathered regolith in this area (Maquaire et al., 2002). In 2019, temperature-probe locations were modified to a single

set of four probes located mid-slope on both the north- and south-facing slope. As our interest is focused on frost weathering, we specifically analysed soil temperatures during the winter season, from October 18th to March 31st. This period was chosen because negative soil temperatures are almost absent outside these dates; During the periods between April 1st and October 17th, we found that the time spent below 0 °C was on average less than 0.4% of the total time in a year and represents less than 4% of the time spent below 0 °C during the winter. The time spent in the frost-cracking window outside of the analysed winter

period was null for most of the study years. We chose to start the winter season on October 18th because some yearly series miss temperature data for early October.

The soil-temperature dataset includes periods of missing or spurious data, due to probe malfunction, unearthing or burial. The relatively mobile nature of the marls leads to frequent displacement of probes, identified in the temperature records by similar temperatures between probes at different depths or anomalous daily temperature ranges (Fig. 2). When these periods are longer

than twenty consecutive days, the full season of data is rejected (i.e., 2005/2006, 2010/2011 and 2012/2013). When the missing period is shorter (for seasons 2011/2012, 2014/2015, 2015/2016), we searched for a relation between soil temperature and air temperature (recorded at the closest weather station) in order to reconstruct soil temperature during the missing time interval (Suppl. Figs. 2 and 3). Over the eleven winter seasons used to calculate our temperature indicators, we reconstructed 56 days in this manner, which represents around 3% of the total winter temperature dataset (see Supplementary Table 1 for details).

We found that a simple two-tier linear fit predicted soil temperature well for air temperature below 10 °C and soil temperature below 5 °C, with an inflexion point around -4°C. The correlation breaks down for higher temperatures due to radiative heating effects; however, since our focus is on soil temperatures below 0 °C, the simple linear-correlation approach is sufficient. The presence of snow sets the soil temperature to ~0 °C (Fig. 2) and thus perturbs the linear trend. Therefore, the regression was calibrated during snow-free periods only.

The soil temperature data has been recorded in the Moulin catchment but is regarded representative also of the adjacent Laval catchment that shares the same lithology. Sediment-yield data is available for both the Moulin and Laval stations (located 100 meters apart at the outlets of both catchments; Fig. 1C) and our analysis of hysteresis cycles shows that both catchments have the same sediment dynamics (see Section 4.1, Figs. 4, 5). Because of its reduced vegetation cover and larger area (0.86 km$^{2)}$, we consider that the Laval catchment is more representative for the analysis of the relationship between sediment yield and

temperature indicators (Figs. 3, 4 and 6 to 9) than the much smaller Moulin catchment (0.089 km$^2$). Moreover, the larger sediment export values from the Laval catchment are associated with smaller relative uncertainties. For these reasons, we focus our attention on the sediment-export data from the Laval catchment for our analysis.

## 3.2 Data Processing

### 3.2.1 Rainfall and sediment export

First, we compared monthly sediment export and monthly rainfall to understand the seasonal dynamics of sediment transport in the Laval and Moulin catchments (Figs. 3 - 5). The analysis was performed for the period 2003 - 2020, during which sediment export was precisely recorded. In this time interval, rainfall amounts were summed for each month to obtain monthly rainfall. When gaps were present in the data (around 6% of the time), daily cumulative reconstitutions of these missing periods were possible thanks to the network of tipping-bucket rain gauges installed around the catchment. Monthly averaged rainfall

intensity was also computed by averaging non-zero values of 5-minute constant time-step rainfall intensities. As established in previous studies (Mathys et al., 2003; Bechet et al., 2016), sediment export at the event scale is driven by rainfall intensity above a threshold. Therefore, we analysed the correlation between annual sediment export and annual rainfall, looking not only at total annual rainfall, but also at annual rainfall above several threshold values of intensity varying from 10 to 80 mm/h. For this correlation, we used the instantaneous intensities computed at the variable time steps of each tipping event. We selected

this range of thresholds based on inferences from previous studies (Mathys, 2006) and then used the value providing the best correlation to predict annual sediment-export values. In order to overlap with the period where soil-temperature data were available, we used only data between 2005 and 2020 for the correlation between rainfall and annual sediment export. We define the sediment-export anomaly as the residual between measured sediment export in any year and the predicted value from this regression. With the aim of quantifying the impact of frost weathering on sediment production, and of identifying

the most relevant frost-weathering process, our objective is to identify a controlling factor to explain this sediment-export anomaly.

### 3.2.2 Temperature indicators

With the aim of quantifying the impact of frost weathering on sediment production and of identifying the most relevant frost-weathering process, we selected four potential indicators of frost-weathering intensity and correlated them both with each other

and with the sediment-export anomaly.

First, we computed the Frost Cracking Intensity Indicator (FCII) based on the model of Hales and Roering (2007), inspired by the segregation ice-growth hypothesis of Hallet et al. (1991). Because reliable data on soil moisture are very challenging to obtain in marly badlands, we considered, based on the work by Mallet et al. (2018), that the marls fulfil the moisture conditions needed for ice-lens growth during winter. Since we had access to direct soil-temperature measurements at different depths, we

used these rather than the diffusive model based on air-temperature developed to predict soil temperatures in the original formulation of Hales and Roering (2007). The soil-temperature gradient was computed between the probes located at 1 cm and 24 cm depth. We summed temperature gradients during all time intervals when the surface temperature (probe at 1-cm depth) was within the frost-cracking window (between -3 and -8 °C) over a winter cycle, and used this number as a proxy for

the frost-cracking intensity. Following Anderson et al. (2013), we express the FCII in units of temperature-time/length (ºC
min/cm in our case). The probes at 24 cm depth on the north-facing slope did not produce reliable data for almost all years
considered; we therefore only computed FCII for the south-facing slope.

The second indicator counts the number of freeze/thaw cycles as a proxy for the efficiency of frost weathering by volume
expansion. The raw temperature data include many insignificant temperature fluctuations around 0 °C, over very short periods
that do not allow the formation of frost. We therefore applied a time threshold of 1 hour below 0 °C to count a freeze/thaw
cycle. The same duration was considered as a threshold for temperature remaining above 0 °C to distinguish two freezing
periods. We observed that increasing this threshold to more than 1 hour did not affect the number of freeze/thaw cycles
significantly; above this threshold the number of cycles tended toward counting the daily temperature cycles.

The third indicator is the time spent at negative temperatures. We counted the number of 10-minute periods with negative
temperature recordings at the 1 cm-depth probe and converted these into hours. Finally, the fourth indicator is the mean
negative temperature, which was computed as the mean temperature value when it is below zero, again for the 1 cm-depth
probe. The behaviours of these indicators were found to be very different between south-facing and north-facing sites, whereas
they did not vary much between uphill and downhill sites. Therefore, we analysed them by averaging the value of the uphill
and downhill indicators for each slope aspect. Fort the 2019-2020 winter season, only a single set of data per slope aspect was
available, measured at a mid-slope location, and we use this in addition to the averaged temperature data from before 2019.

### 3.2.3. Uncertainties

We accounted for uncertainty in our analyses using the following procedures. We defined the uncertainty on the sediment-
export anomaly resulting from the linear regression between rainfall and sediment export as the 2-$\sigma$ error on the sediment-
export values predicted by the regression. Whereas the sediment-export and rainfall measurements are associated with
uncertainties themselves, these are insignificant with respect to the uncertainty on the regression and were therefore not
included in the analysis. For the temperature indicators, we defined the uncertainty around their mean value by the difference
between the value of the indicator obtained at the uphill and downhill location on the same slope (i.e., north-facing versus
south-facing).

To establish a relation between sediment-export anomalies and temperature indicators while accounting for these uncertainties
in both parameters, we used weighted linear regression with the uncertainty in both independent and dependent variables as
(inverse) weights (cf. York et al., 2004). Since a conventional $R^2$ is invalid in the presence of measurement errors (Cheng et
al., 2014), we characterise the quality of the weighted regressions by a weighted correlation coefficient $R^2_w$ computed from
the weighted residual ($Var_{res-w}$) and weighted total ($Var_{tot-w}$) variance ($R^2_w = 1 - (Var_{res-w} / Var_{tot-w})$). The goodness-of-fit of the
weighted correlation is expressed by the weighted sum of deviations from the best-fit line S, normalised by the degrees of
freedom (n-2): $S_n = S / (n-2)$ (York et al., 2004). We also computed standard $R^2$ and p-values from an ordinary least-squares

linear regression as these are more readily interpreted in terms of significance of the regression. Because of the apparent differences in correlations between the north-facing and south-facing slopes, we analysed correlations for both slope aspects separately.

## 4. Results

### 4.1. Seasonal variability in sediment export

Box plots of monthly precipitation, monthly rainfall intensity, and total sediment export (i.e., bedload and suspended load) recorded between 2003 and 2020 show high disparity between monthly averages and significant interannual variation for each month (Fig. 3). Median monthly rainfall values are low in winter and summer and peak in spring and autumn, varying from ~50 mm in January-March to ~100 mm in October-November, with extreme values reaching ~300 mm in those months (Fig. 3A). In contrast, the monthly average rainfall intensity (Fig.3B) shows a single peak in summer, with median values varying between ~2.7 mm/h in January up to ~7.5 mm/h in July. Monthly total sediment export also shows two peaks (late spring and autumn), with minimum and maximum values of median sediment export around 0 and 3.6 ktons in Januaryand June, respectively. The variability in monthly total sediment export is high and extreme values reach almost 10 ktons in July.

For all three datasets, (i.e., rainfall, rainfall intensity and sediment export), higher monthly values are associated with higher variability (i.e., values for November, July and June, respectively). For total sediment export, most months show outlier values and distributions are skewed toward high values (mean > median). Rainfall boxplots show a larger spread between the first and third percentiles but more symmetric distributions.

Despite noticeable disparities between years (Fig. 3, Suppl. Fig. 4), mean values highlight a strong seasonal pattern in precipitation, rainfall intensity and sediment export, as previously observed (Mathys et al., 2003; Fig 4). In the Laval catchment, sediment export is low in winter and increases in spring to a peak in June, with an average monthly export of ~3.2 ktons that can be associated with a significant increase in average monthly rainfall intensity (~6 mm/h) (Figs 3B, 4B), despite relatively low total rainfall during that month (<80 mm). Conversely, autumn months (October, November) are characterized by high total rainfall (>100 mm/month) but much lower rainfall intensity (<4.5 mm/h) and sediment export (<1.5 ktons/month). Thus, rainfall intensity appears to play a key role in triggering spring sediment export . Despite their difference in size, the monthly distribution of total sediment export is similar between the Moulin and Laval catchments (Fig. 5): both show a major peak in June, amounting to ~20% of total yearly sediment export, with smaller peaks (~10%) in August and October. The monthly rainfall distribution shows a main peak in October-November, with a secondary peak in May. The spring rainfall peak (May) occurs one month earlier than the sediment-export peak (June) for both catchments.

Based on annual data records since 2005 and previous work (see Methods section), we established a correlation between rainfall intensity above a threshold and sediment export (Fig. 6). An optimal correlation is obtained for a threshold of 50 mm/h ($R^2 = 0.86$), with most scatter occurring for cumulative rainfall (above threshold) values between 0 and 40 mm. This correlation

was used to define the sediment-export anomaly as the residual between observed and predicted annual sediment-export values for each year.

## 4.2. Potential temperature control on sediment export

Daily temperature fluctuations in the bare marly soils that were monitored are significantly higher for the shallow (-1 cm) than for the deep (-24 cm) sensors (Fig. 2). During summer, surface temperatures show a daily variability of ~40 °C (between ~10-15 °C at night and ~50-55 °C at midday), whereas the daily variability in winter averages around 20 to 30 °C depending on aspect. These daily variabilities decrease drastically with depth, fluctuating generally within 5 °C and 20 °C for the probes at -24 cm and -12 cm depth respectively, but can occasionally be higher. The maximum daily temperature is reached asynchronously according to the exposure; in the north-facing hillslope, maximum temperature occurs in the morning (around 10 am), whereas in the south-facing slope it occurs late in the afternoon (around 4 pm). At the maximum depth (-24 cm), minimum soil temperatures during winter are negative on the north-facing slope but positive on the south-facing slope.

We calculated four different temperature indicators during the winter months, as explained in section 3.2.2, and compared these with each other as well as with the sediment-export anomaly obtained previously, in order to (1) assess the degree of co-variation between the different indicators, and (2) characterize the direction and strength of the relationship between the temperature indicators and the sediment-export anomaly. Overall, stronger correlations were found between temperature indicators on the south-facing slope than on the north-facing slope (Fig. 7). Frost-cracking intensity, which was only computed on the south-facing slope, correlates strongly ($|R| > 0.67$) with the other temperature indicators. Time below 0 °C is also strongly correlated with the number of freeze-thaw cycles / year ($R = 0.90$) on the south-facing slope. All of the correlations between temperature indicators are significant ($p < 0.05$) on the south-facing slope. On the north-facing slope, in contrast, no significant correlations between temperature indicators were found. Sediment-export anomalies correlate most strongly with frost-cracking intensity on the south-facing slopes ($R = 0.87$) but are also significantly correlated with time below 0 °C ($R = 0.77$). On the north-facing slope, the only significant correlation occurs between sediment-export anomaly and time below 0 °C ($R = 0.75$), but note that frost-cracking intensity was not calculated on the north-facing slope.

These correlation results led us to investigate how much of the variability in sediment-export anomalies can be explained by these different temperature indicators, taking into account the uncertainties on both measures (Figs. 8, 9). Results show that the weighted regression between frost-cracking intensity and sediment-export anomalies can explain 47% of the variance in the latter ($R_w^2 = 0.52$). The ordinary least-squares (OLS; i.e., unweighted) correlation is significant ($p = 0.005$) and only slightly deviates from the weighted regression (Fig. 8). Based on this trend, the highest sediment-export anomalies appear to occur in years succeeding winters with strong (negative) frost-cracking intensity.

A significant positive relationship is also found between sediment-export anomalies and time below 0 ºC on both the south-facing ($R_w^2 = 0.77$) and north-facing ($R_w^2 = 0.51$) slopes (Fig. 9). In contrast, regressions between sediment-export anomalies

and either the number of freeze-thaw cycles / year or the mean negative temperature are not significant on either slope and show an opposite trend between slopes (i.e., a positive correlation on one slope and negative on the other; Fig. 9). This analysis suggests that both the frost-cracking intensity (where it can be calculated) and the time spent below 0 ℃ in a particular winter, which are strongly correlated (Fig. 7, Suppl. Fig. 5) are good predictors of sediment-export anomalies (deviations from sediment export predicted by rainfall over a threshold) in the following year.

## 5. Discussion

### 5.1. Limitations of our study

Because of the strong annual and inter-annual variability in regolith cover and sediment export, long-term field measurements at high spatial and temporal resolution are required to characterise the dynamics of badland erosion. The Draix-Bléone CZO provides one of the few localities worldwide where such records exist. Similar datasets have been collected in the Araguas and Vallcebre basins of (Northern Spain) but cover significantly shorter periods (6-18 months; Regüés et al, 1995; Regüés and Nadal-Romero, 2013). However, uncertainties in both sediment-yield records and soil-temperature measurements are inevitable and difficult to estimate.

The interpretation of the hysteresis cycle (Fig. 4) should integrate the variability in sediment export (Fig. 3C). Although June shows the highest sediment export overall, this month is also characterised by the highest inter-annual variability in sediment export. This variability can be explained by the stochastic nature of precipitation, which occurs mainly due to storms during the spring and summer seasons. Thus, sediment-export values for June vary between 0 (no large rainfall events during the month) up to 8000 tons. The monthly data show that the highest sediment export can also occur in July (and more rarely in August) in some years. In order to smooth out such stochastic behaviour, we integrated sediment-export values over the full year following each investigated winter season.

The export-anomaly values that we computed are dependent on the linear regression with rainfall above a threshold (Section 4.1; Fig. 6). Extreme values, such as the three years with >70 mm or more rainfall above the 50 mm/h threshold, have an important impact on the regression and thus on the export-anomaly values. Calculating uncertainties on the annual sediment export is challenging, but an order of magnitude of around 10% of the total sediment export can be estimated. This uncertainty is negligible compared to the uncertainties on sediment–export anomalies that we infer from the regression analysis. The uncertainty on annual rainfall is also considered negligible.

Soil-temperature probes have proven very difficult to maintain at the depths where they were installed in the soft, mobile marls of our study area. For this reason, we could only calculate frost-cracking intensity (which requires concomitant data from the -1 cm- and -24 cm-depth probes) for the south-facing slope and for only 7 years between 2005 and 2019 (Fig. 8). We therefore searched for a temperature proxy that allows predicting frost-weathering intensity with less constraints on the data, and found that the time spent below 0 ℃ (for the -1 cm-depth probe) can be a useful indicator.

When assessing the predictive power of the different temperature indicators to explain sediment-export anomalies, we aimed to take into account the uncertainties in both variables (Figs. 8, 9) by employing weighted regression based on uncertainties (York et al., 2004). However, the uncertainties in temperature indicator values were computed from only two points, i.e., the measurements at the uphill and downhill locations of each slope aspect. Uncertainties on both variables should also be of the same order of magnitude to avoid biasing the weighted regression, whereas in our case, the relative uncertainties in the

temperature indicators can be much larger than those in the sediment-export anomalies. For this reason, we also report the ordinary least-squares regression and associated significance (p-value).

## 5.2. Significance of the observed hysteresis cycle

Numerous studies have reported annual hysteresis cycles between rainfall or discharge on one hand, and sediment export on

the other; these studies have commonly focused on large catchments, e.g. in the Andes or Himalaya (e.g., Andermann et al., 2012; Armijos et al., 2013; Tolorza et al., 2014; Li et al., 2021). For such large catchments, the annual hysteresis cycle is explained by the role of subsurface water storage (Andermann et al., 2012), dilution effects (Armijos at al., 2013) or variations in the contributive erosive area (Li et al., 2021). Hysteresis cycles for smaller catchments have generally been analyzed at the event scale, and have been interpreted in terms of the proximity of sediment sources and the spatio-temporal heterogeneity of

rainfall (e.g., Klein, 1984; Buendia et al., 2016). More directly comparable to our results, several studies have been carried out in small (< 15 km²) Mediterranean badland catchments where climate can vary between arid to humid conditions. Llena et al. (2021) reported a seasonal sediment dynamic with lags between sediment production and sediment yield and highlighted the role of the channel network in the sediment transfer. Several catchments in Northern Spain that are very similar in size and lithology to our study site have been well studied and suspended sediment transport processes have been reconstructed at the

event (Soler et al., 2008; Nadal-Romero et al., 2008) and seasonal scale (Nadal-Romero and Regüés, 2010). Counter-clockwise and clockwise hysteresis loops in these catchments are associated to dry and wet seasons, respectively, and are inferred to be driven by infiltration and saturation processes on hillslopes.

The annual hysteresis cycle between rainfall amount and sediment export observed in the Draix-Bléone CZO (Fig. 4A) presents two loops with successively anti-clockwise and clockwise patterns, reflecting the rapid seasonal changes in erosion regime in

these badlands. We interpret the initial anti-clockwise hysteresis loop, with sediment export lagging behind precipitation in the first half of the year, to be due to the threshold in rainfall intensity required to generate erosion. The analysis of the annual hysteresis cycle between monthly average rainfall intensity and sediment export confirmed this interpretation; it is only with the high-intensity storms of late spring/early summer (June) that significant amounts of sediment are exported from the catchment (Fig. 4B). The initial anti-clockwise hysteresis loop of Fig.4A thus indicates transport-limited conditions. In

contrast, the clockwise loop characterising the second half of the year indicates supply-limited conditions, with the supply of mobile sediments running out after summer (Bechet et al., 2016). The hysteresis cycle between average rainfall intensity and

sediment export highlights this lack of sediment supply during the summer: the mean rainfall intensity peak (July) lags behind sediment- export peak (June) and sediment export between July and October is steady at a lower value than the first half of the year (around 1.5 ktons per month), independently of rainfall intensity. A similar trend appears in the Moulin catchment, despite its much smaller catchment area (Fig. 5). This seasonal pattern does not seem to change with recent climatic variations because it was already reported by Mathys et al. (2003), who analysed the seasonality of bedload yield based on Draix-Bléone observatory data between 1985 and 2003.

The annual pattern that we describe in Fig. 4A characterizes the total sediment export, which is the sum of the suspended sediment and bedload yield. According to the observations of Mathys et al. (2006) and Liébault, 2017, highest export values for bedload occur during autumn. However, this trend is modified when adding the suspended load because the exported suspended-sediment mass can be up to four times higher than bedload-sediment mass during spring. A similar ratio has been observed in other catchments (Lana-Renault and Regüés, 2007; Rainato et al., 2017) and leads to the present result where late spring/early summer (more specifically, June) is the period that contributes most to the total sediment yield. Thus, sediment dynamics vary according to grain size: intense precipitation events during late spring/early summer mobilise the regolith produced during the winter, and suspended sediments are transported almost directly from the hillslopes to the catchment outlet. In contrast, coarser bedload sediments are deposited in gullies during these short intense events and are only exported by the lower-intensity but longer-duration rainfall events of the late summer and autumn (Figure 4). These dynamics suggest that suspended sediment storage is almost non-existent in the Moulin and Laval catchments, whereas bedload sediment can be stored for several months.

**5.3. Frost weathering as a major control on sediment production in the Draix-Bléone CZO**

The significant correlations between frost-cracking intensity or time below 0 °C and sediment-export anomalies imply that frost-weathering processes constitute a major secondary control on sediment export. While the main control on the yearly amount of sediment exported from the studied catchments is exerted by rainfall above a threshold (Fig. 6), the efficiency of frost-weathering processes during the preceding winter, as expressed by the frost-cracking intensity indicator or the time below 0 °C, can explain about half of the residual from this trend (Figs. 8, 9). Together with the evidence for a transition from transport-limited to supply-limited conditions during the year discussed above, we interpret these results as indicating that frost-weathering processes modulate sediment export from the catchments by exerting a strong control on the production of mobilizable sediment. In particular, the lack of sediment supply during summer months inferred in the previous section argues against significant sediment production by solar-induced thermal stresses (e.g.,Eppes et al., 2016), despite high daytime surface temperatures and large temperature variations in the marls during summer (Suppl. Fig. 1).

These findings are consistent with those of Rengers et al. (2020), who found a strong positive correlation between the time spent in the frost-cracking window and sediment production feeding debris-flow channels on a small, steep plot in the Rocky Mountains of Colorado. The similar results between both studies, despite differing scales, lithologies and geomorphic settings,

attest to the potential widespread control of frost-weathering processes on sediment production. Additionally, our results like

those of Rengers et al. (2020) show a weak correlation between sediment production and the number of freeze-thaw cycles per year. As discussed in the introduction, freeze-thaw cycles are associated with volumetric expansion of ice, whereas frost-cracking is related to ice segregation. Thus, the migration of liquid water to loci of ice-lense growth appears to be a more efficient process of soil weathering than volumetric expansion, even in this temperate and moderate-elevation environment. Nadal-Romero et al. (2007) similarly concluded that frost-weathering processes play an important role in weathering of marly

badlands in northern Spain; they also demonstrated significant variations between north-facing versus south-facing slopes. However, Nadal-Romero et al. (2007) focused their attention on freeze-thaw cycles and did not investigate frost-cracking, the quantification of which requires more data. Nonetheless, our studies concur in underlining the influence of frost-weathering processes on regolith development as well as important spatial variations between north- and south-facing slopes, supporting the theoretical predictions of Anderson et al. (2013).

Directly quantifying frost-cracking intensity (rather than inferring it from atmospheric temperature data) requires dense and high-quality field data, in particular concomitant soil-temperature data at multiple depths to quantify thermal gradients. Even in permanently monitored long-term observatories such as Draix-Bléone, such data may be rare. However, our study suggests that the time spent below 0°C, which correlates well with the frost-cracking intensity (Fig. 7; Suppl. Fig. 5), may be used as a simpler proxy to predict frost-weathering intensity. Our weighted regressions show that this indicator correlates well with the

sediment export anomaly and captures the effect of frost-weathering processes on sediment production, even though uncertainties in the regression can be significant (in particular on the south-facing slope in our case). In contrast, neither the mean negative temperature nor the number of freeze-thaw cycles appear as reliable proxies to estimate frost weathering in this setting.

These correlations between sediment production computed during a winter season and sediment yield for the directly following

year (spring to autumn), together with the strongly varying dynamics of transport during the year discussed in the previous section, favour the hypothesis of rapid sediment export from the studied catchments, in contrast to the 3-year residence time of sediments in these catchments inferred by Jantzi et al. (2017). In order to test this hypothesis, we performed correlations between the frost-cracking intensity in a particular winter season and the sediment-export anomaly of the first, second and third year after that season (i.e., in the last case, if we consider the 2006-2007 winter season for the frost-cracking intensity,

we compare it to the sediment-export anomaly for 2010). In all configurations, the correlation is weaker than the direct annual correlation that we observed ($R^2 = 0.76$) and the correlation weakens with increasing residence time (Suppl. Fig. 6); correlations for the years n+2 and n+3 are not significant. The ratio observed in sediment distribution during the spring/summer (74% suspended load / 26% bedload) and the rapid export of these fine sediments probably make the suspended load invisible in the estimation of sediment storage in the catchment, rendering the calculation of residence time complex. Thus, it appears that the

Laval catchment has an efficient drainage system, with high connectivity and low sediment storage, which does not contribute significantly to the production / export balance. Similar short residence times have been observed, at a different scale, in a similar environment in northern Spain (Andres Lopez-Tarazon et al., 2011).

## 5.4. Implications for the erosional response to climate change

Due to their limited vegetation cover and soft lithology, badlands are sensitive areas that are directly exposed to climatic
parameters and thus will respond quickly to even small climatic variations (e.g., Clarke and Rendell, 2010). In the context of anthropogenic climate change, weathering or erosion processes and hydrology will necessarily be modified: a positive or negative variation of one of these main operating processes could be balanced against others but could also be additive, thereby inducing important changes in morphology and sediment export in these landscapes.

Based on a review of multiple badland areas in the Mediterranean region, Nadal-Romero et al. (2021) have recently
investigated the impact of climate change on these particular landscapes, taking into account different climatic drivers (rainfall amount, rainfall intensity, wetting-drying cycles, freeze-thaw cycles, soil moisture content, etc.). Their analysis predicts that for wet badlands such as Draix-Bleone, the expected increase in rainfall intensity should increase erosion capacity on one hand, but that the expected increase in temperature should lower the number of freeze-thaw cycles, thereby reducing the efficacy of frost-weathering processes and decreasing sediment availability on the other hand. Considering the result from our
study that frost-cracking intensity, and not freeze-thaw cycles, is the best indicator for sediment production by frost-weathering processes, the frost intensity rather than the number of frost days should present a better proxy to predict the evolution of sediment availability. However, the time spent below 0°C, identified here as a simple alternative proxy for frost-weathering intensity, will also be directly affected by an increase in temperature and similarly lead to a decrease in sediment production in a warming climate.

Recently, Hirschberg et al. (2021) have also shown that projected changes in precipitation and air temperature would lead to a reduction in both sediment yield and debris-flow activity in an Alpine catchment at moderate elevation (< 2000 m), because of the reduction in frost-weathering intensity. In general, expected changes in sediment production in temperate regions under a warming climate appear to have a counter-effect to the predicted increase in average rainfall amounts and intensities, which have been considered as "the most direct factors controlling erosional changes under climate changes" (Nearing et al., 2004).
This complex interaction between sediment production and sediment transport underlines the necessity to account for the processes responsible for sediment production in longer-term predictions of sediment yield.

Finally, in addition to physical weathering processes that are discussed here, chemical and biological weathering processes also play an important role in rock weathering and will also be affected by precipitation and temperature changes (e.g., Brantley et al., 2011; Soulet et al., 2021). Additionally, erosion processes could be impacted by changes in the vegetation but these
interactions are particularly complex to understand (Nearing et al., 2004). Therefore, climatic variations may change the balance between weathering and erosional processes as well as their timing, leading to complex positive or negative feedbacks on catchment erosion that remain difficult to predict.

## 6. Conclusions

Based on our analysis of sediment-yield records and soil-temperature data from the Draix-Bléone CZO and accounting for the inevitable uncertainties in our dataset, we show that frost-weathering processes modulate sediment export by controlling sediment production in these marly catchments. Our main conclusions are summarized below:

- Monthly total sediment export (suspended load and bedload) is highly variable and shows a seasonal pattern. The annual hysteresis cycle (Fig. 4) shows an anti-clockwise pattern in the first half of the year (February-July),and a clockwise pattern later in the year (August-December), suggesting a spring / early summer transport-limited regime followed by a supply-limited regime during late summer and autumn in these catchments.

- Total annual sediment export is well correlated with rainfall above an intensity threshold; a threshold of 50 mm/h maximises the correlation (Fig. 6).

- The frost-cracking intensity indicator, calculated following the Hales and Roering (2007) model, explains about half of the sediment-export anomaly (Fig. 7), implying that the process of frost weathering through ice segregation strongly controls regolith production and modulates sediment export.

- The time spent below 0°C is an easier to measure and simpler proxy, and is also an indicator of frost-weathering intensity that correlates well with the sediment export anomaly (Fig. 7). In contrast, neither the number of freeze-thaw cycles nor the average negative temperature during a winter season show significant correlations with sediment-export anomalies.

- South- and north-facing slopes show distinct behaviour with respect to frost-weathering processes, confirming observations from previous studies.

- Frost-weathering processes should be taken into account when building predictive models of sediment export under a changing climate. Under a warming climate, frost weathering should become less important, counteracting the increased sediment export expected from a stormier climate.

The long-term monitoring records available for the Draix-Bléone CZO catchments have allowed identifying frost-cracking by ice segregation as a major control on sediment production. Further assessment of the importance of this process would require similar studies in different environments (e.g., drier climates, more vegetated areas, higher elevation, etc.) and at different catchments scales. Field measurements of sediment export and soil-temperature are ongoing in the Draix-Bléone observatory and additional data could be incorporated to the present results in the next few years. In a context of global climate change, future measurements might illustrate the consequences of climate variations at human timescales on processes in the critical zone.

**Data Availability**

The sediment-yield data used in this manuscript is available on BDOH data portal: https://bdoh.irstea.fr/DRAIX/ and referenced under doi:10.17180/OBS.DRAIX.

**Acknowledgements**

This research has been supported by the French National Research Agency (ANR) under the grant ANR-18-CE01-0019-01 (DEAR project) and INRAE Grenoble. This study was carried out in the Draix-Bléone Observatory (France) and used its infrastructure and data. The Draix-Bléone Observatory is funded by INRAE (National Research Institute for Agriculture, Food and Environment), INSU (National Institute of Sciences of the Universe) and OSUG (Grenoble Observatory of Sciences of the Universe) and is part of the French network of Critical Zone Observatories OZCAR, which is supported by the French

Ministry of Research, French Research Institutions and Universities. Comments by two anonymous reviewers helped improving this manuscript.

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

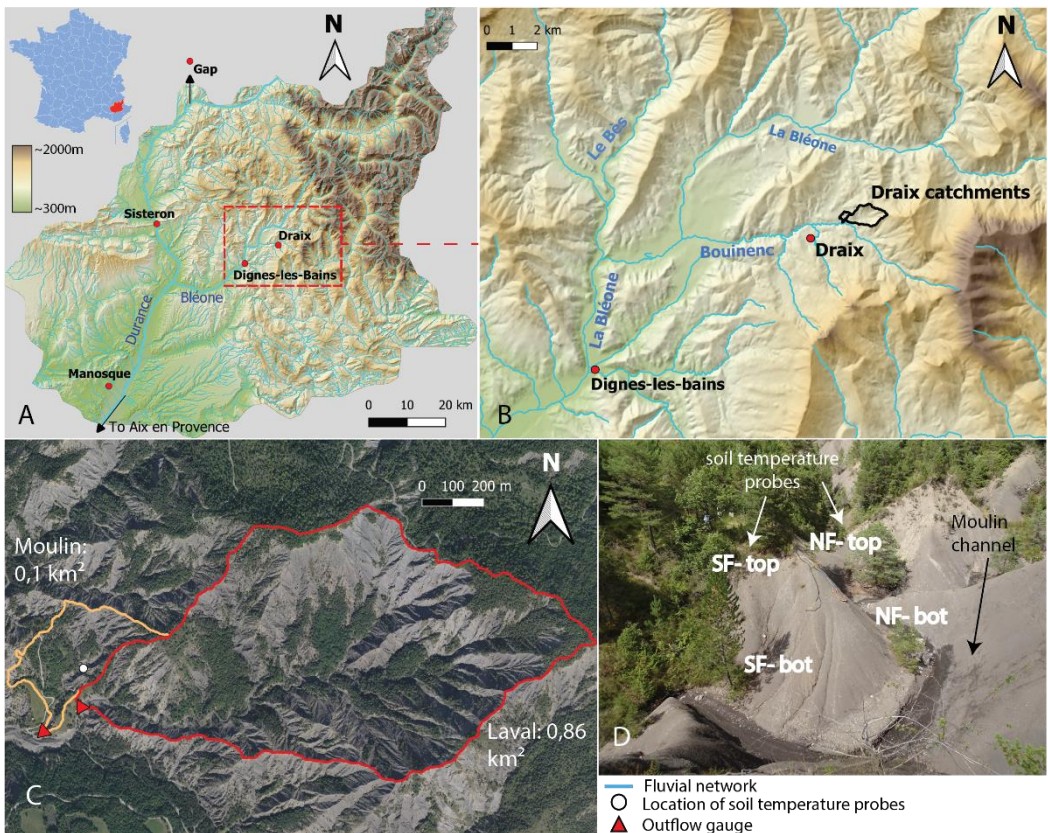

**Figure 1: Geographic setting of Draix-Bléone Critical Zone Observatory. (A) Location within the 'Alpes de Hautes-Provence' department of southeast France (inset), red box shows location of B. Maps are from DEM RGEALTI IGN database; (B) Zoom on the Bouinenc catchment, black outlines show Moulin and Laval catchments. (C) Satellite image (Géoportail) of the Moulin and Laval catchments, showing location of outflow gauges and soil-temperature probes. (D) Setting of soil-temperature measurement sites, in a meander of the Moulin stream on north-facing (NF) and south-facing (SF) slopes at different locations (uphill and downhill) on the hillslopes.**

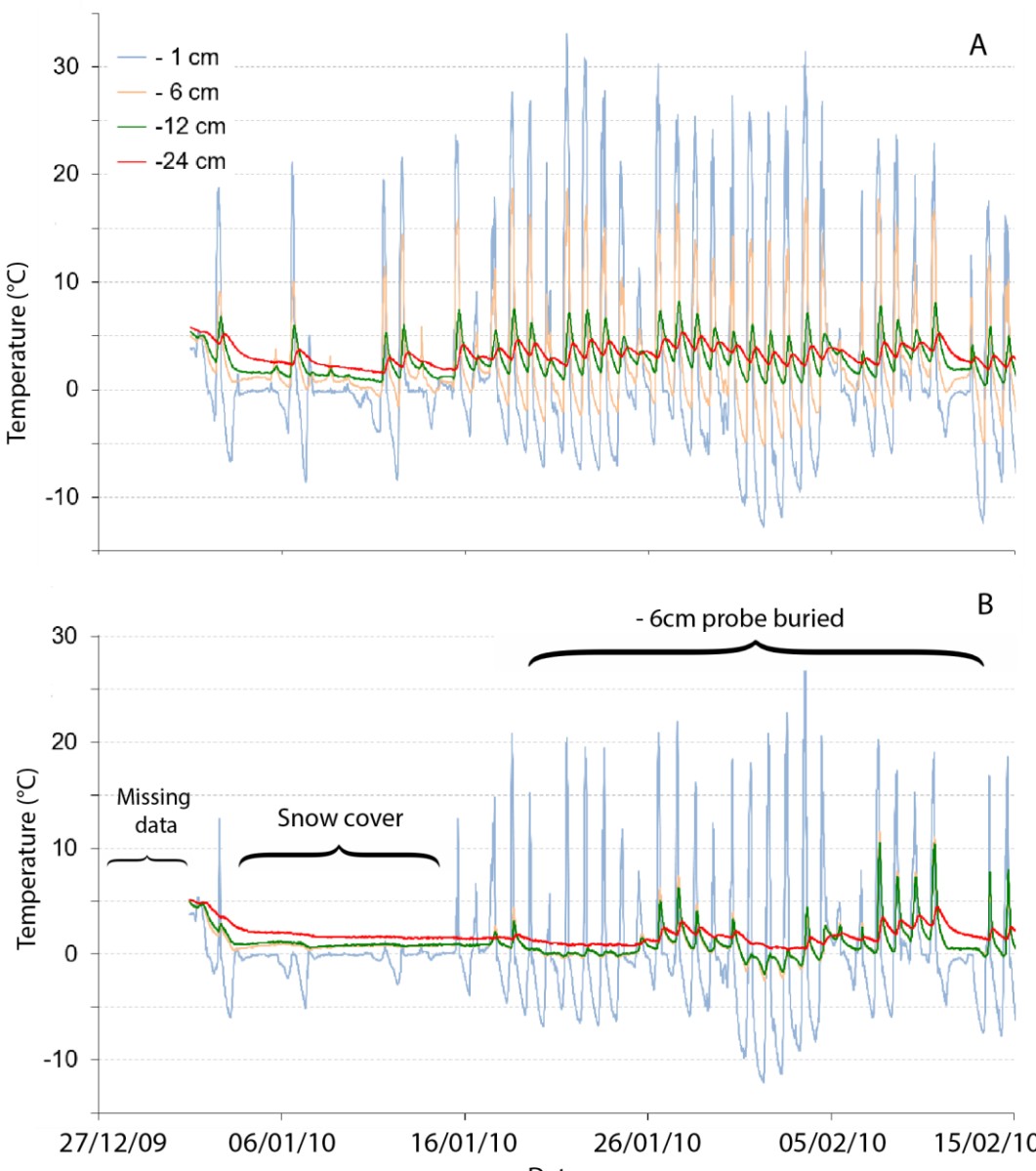

Figure 2: Example of raw temperature series (1 measurement every 10 min). (A) Typical soil-temperature series recorded with four probes at different depths (from south-facing uphill location; time scale as in B). (B) Example of soil-temperature series (from south-facing downhill location) biased because of climatic conditions (snow cover), buried or loosened probes. A full year of temperature measurements is shown in Supplementary Figure 1. High temperature values are observed at -1 cm even in winter when the black marls heat radiatively during sunny periods.



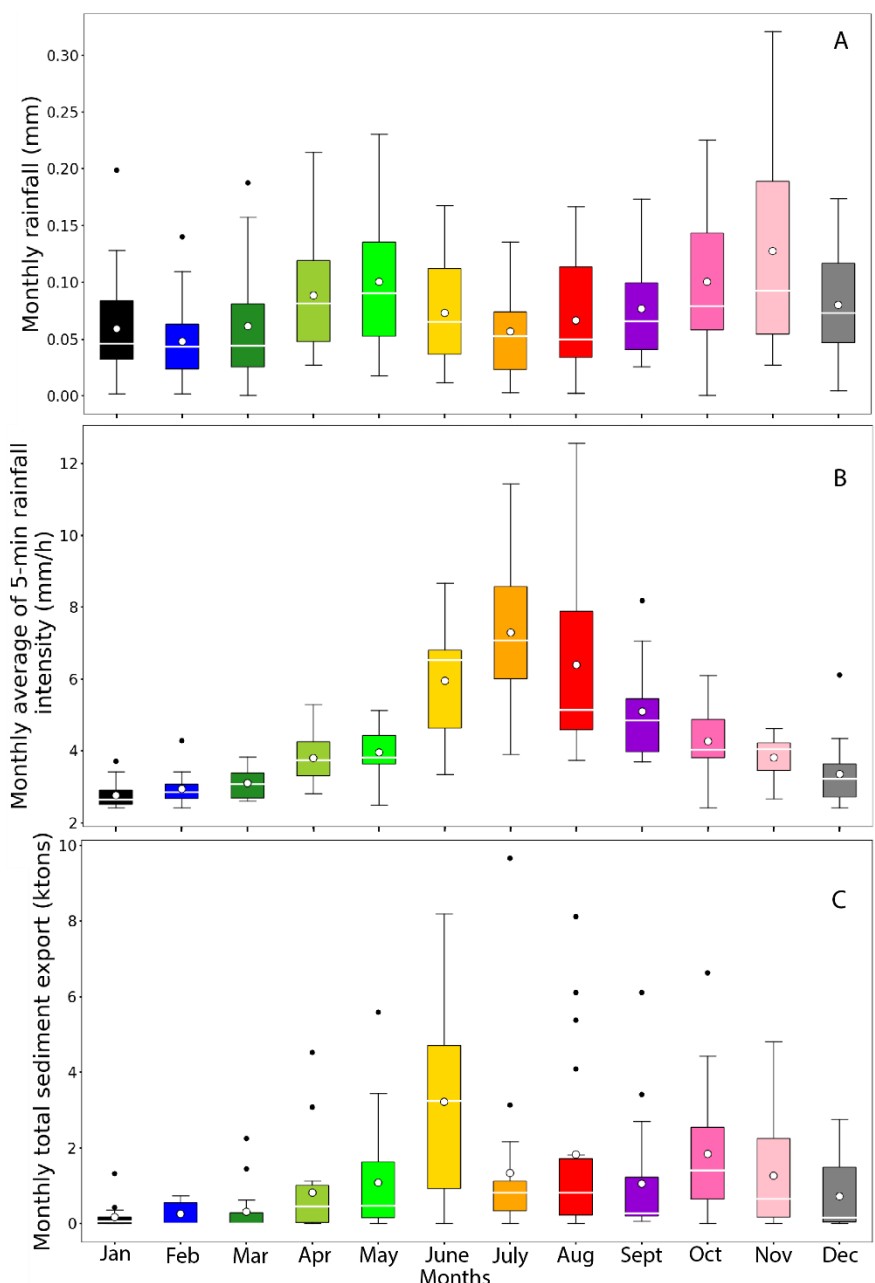

**Figure 3: Boxplots of (A) monthly rainfall, B) Monthly average rainfall intensity computed at 5-minute time-steps and C) monthly total sediment export (i.e., bedload + suspended load) of the Laval catchment. White lines show median values and white dots indicate**

**mean values. The first (Q1, 25%) and third (Q3 75%) percentiles are indicated by the box limits, whiskers show Q1 – 1.5\* IQR and Q3 +1.5\*IQR (inter-quartile range). Black dots are outlier values (i.e., with values <Q1 - 1.5 IQR or > Q3 + 1.5 IQR).**

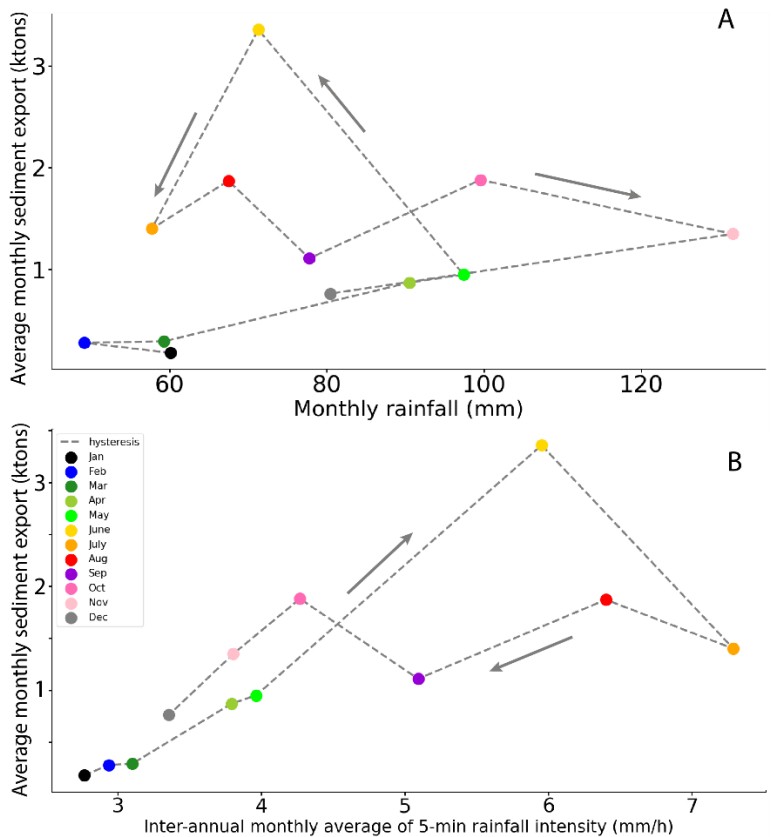


**Figure 4 Hysteresis plot using interannual monthly average values from the Laval catchment between 2003 and 2020. (A) Monthly sediment export (ktons) versus monthly rainfall. Dashed line highlights the hysteresis cycle with two separate maxima: high sediment export and moderate rainfall in June versus high total rainfall and moderate sediment export in October/November (B) Monthly sediment export (ktons) versus monthly average rainfall intensity. Dashed**

**line illustrate the hysteresis cycle with a maxima of sediment export in June preceding a stationary period with around 1.5 ktons of sediment export per month between July and November. A plot of all monthly rainfall averages is shown in Supplementary Figure 4.**

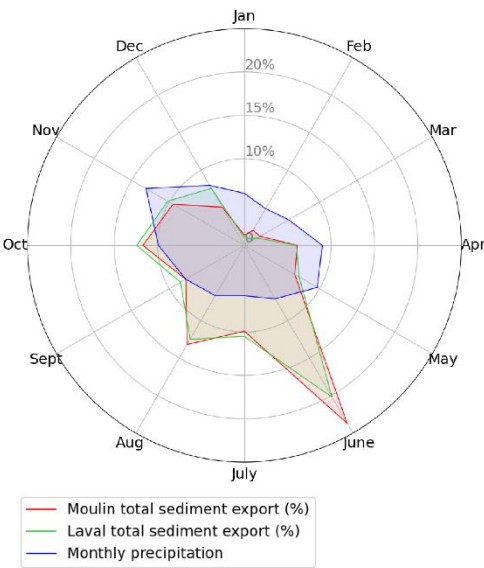

**Figure 5: Comparison of the relative distribution of monthly sediment export for the Laval and Moulin catchments, together with relative monthly rainfall distribution.**

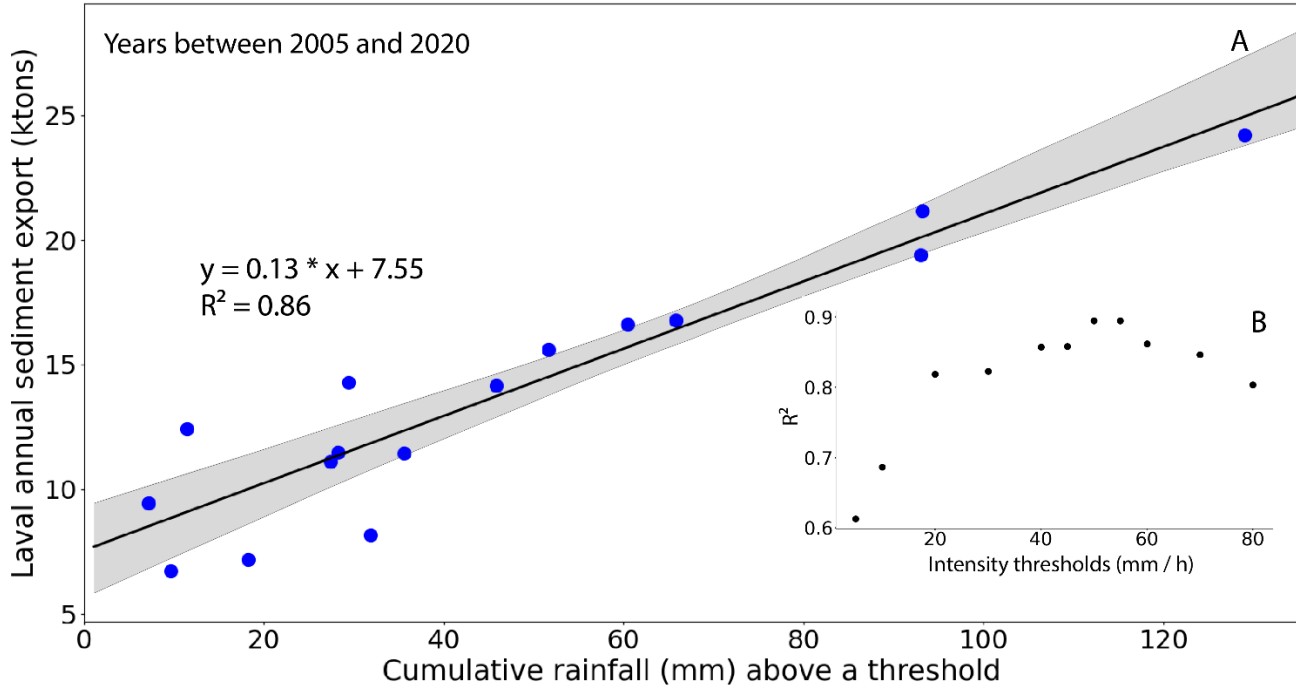

**Figure 6: (A)** Linear correlation between annual total sediment export from the Laval catchment and the cumulative rainfall above an instantaneous intensity threshold of 50 mm/h for the years 2005 to 2020 (blue dots). Regression line is in black; grey shaded area shows 95% confidence interval. Most outliers occur for low cumulative rainfall above the threshold (< 40 mm). **(B)** Coefficient of determination ($R^2$) between annual sediment export and annual rainfall above threshold for different intensity thresholds. Optimum correlations are found for threshold values between 50 and 55 mm/h.

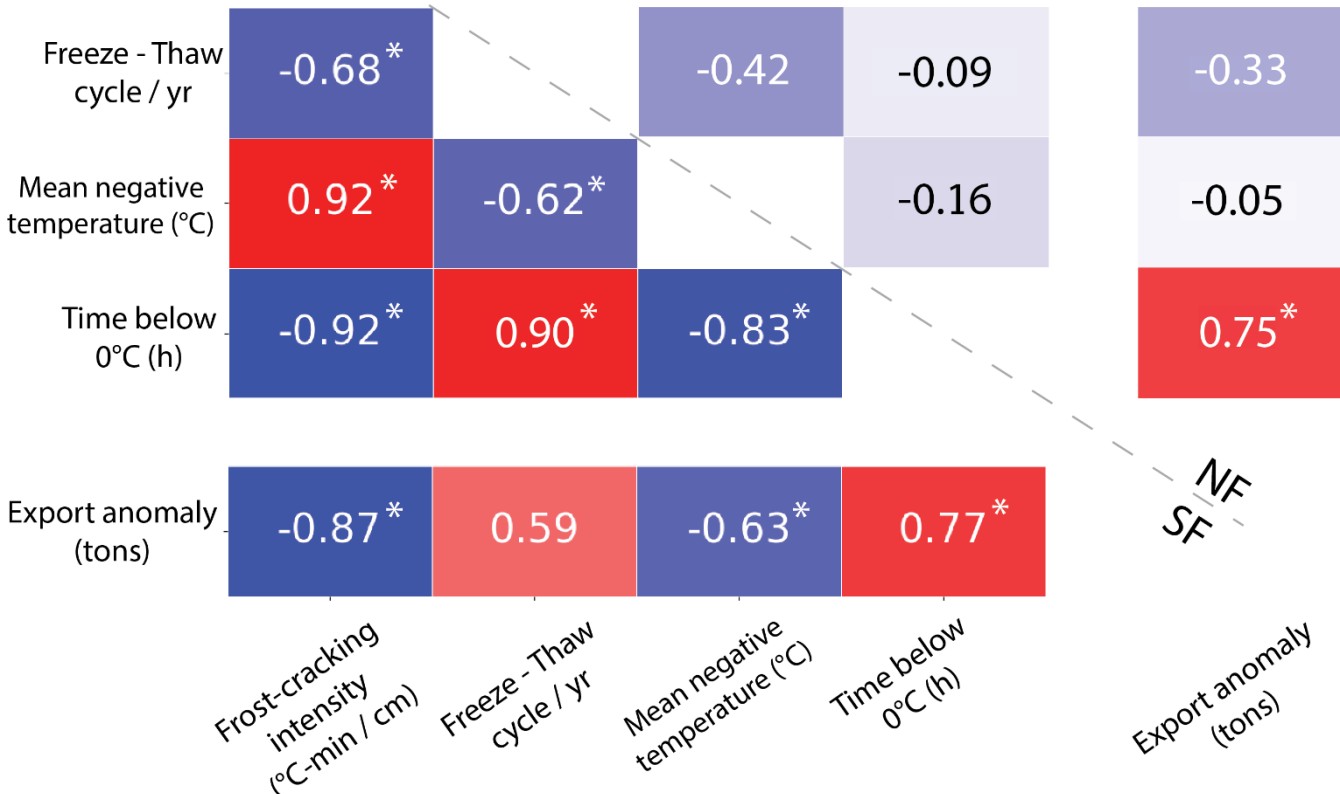

**Figure 7: Correlation matrix between different soil-temperature indicators computed from all data points (using both uphill and downhill locations on the same slope); and between soil-temperature indicators and sediment-export anomaly (calculated using averages between uphill and downhill locations from the same slope). Lower-left and upper-right parts of the table report correlations for south-facing (SF) and north-facing (NF) slopes, respectively (note that frost-cracking intensity was not calculated**

**for north-facing slopes; see text). Blue boxes show negative correlations and red boxes show positive correlations; values indicate correlation coefficient R and stars (*) indicate significant correlations (i.e., $p < 0.05$).**

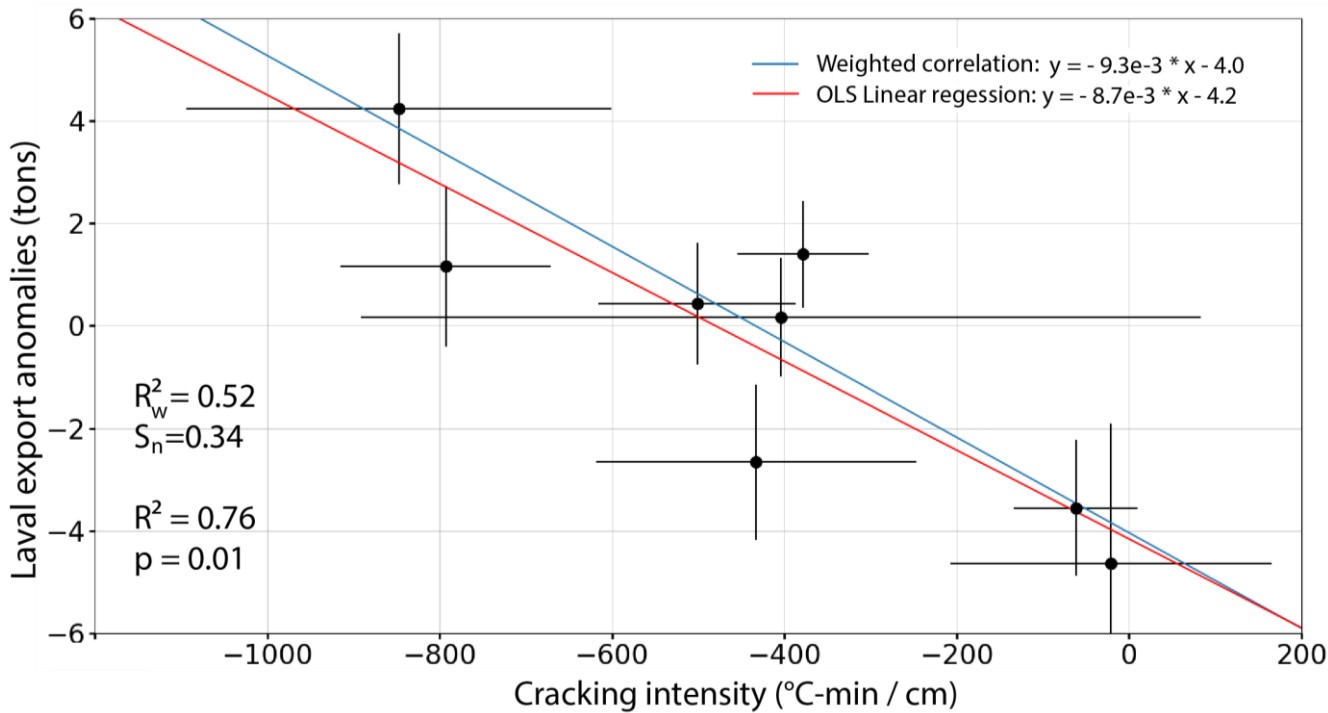

**Figure 8: Regression analysis between sediment-export anomalies and frost-cracking intensity on the south-facing slope for the years 2007 - 2010, 2014, 2015, 2017 and 2020. Horizontal error bars refer to the difference between measurements at uphill and downhill locations (temperature measurements for 2017 were only available for the downhill location, thus uncertainty on frost-cracking intensity was computed as the average of the uncertainties from the others years); vertical error bars are ±2σ uncertainty in export anomaly. Red line shows ordinary least-squares (OLS) linear regression; blue line shows weighted linear regression following York et al. (2004). Weighted determination coefficients $R^2_w$ and associated normalized goodness-of-fit indicator $S_n = S / (n-2)$ are indicated for the weighted regression. Standard $R^2$ and associated p-value indicate significance of the ordinary least-squares (unweighted) regression.**

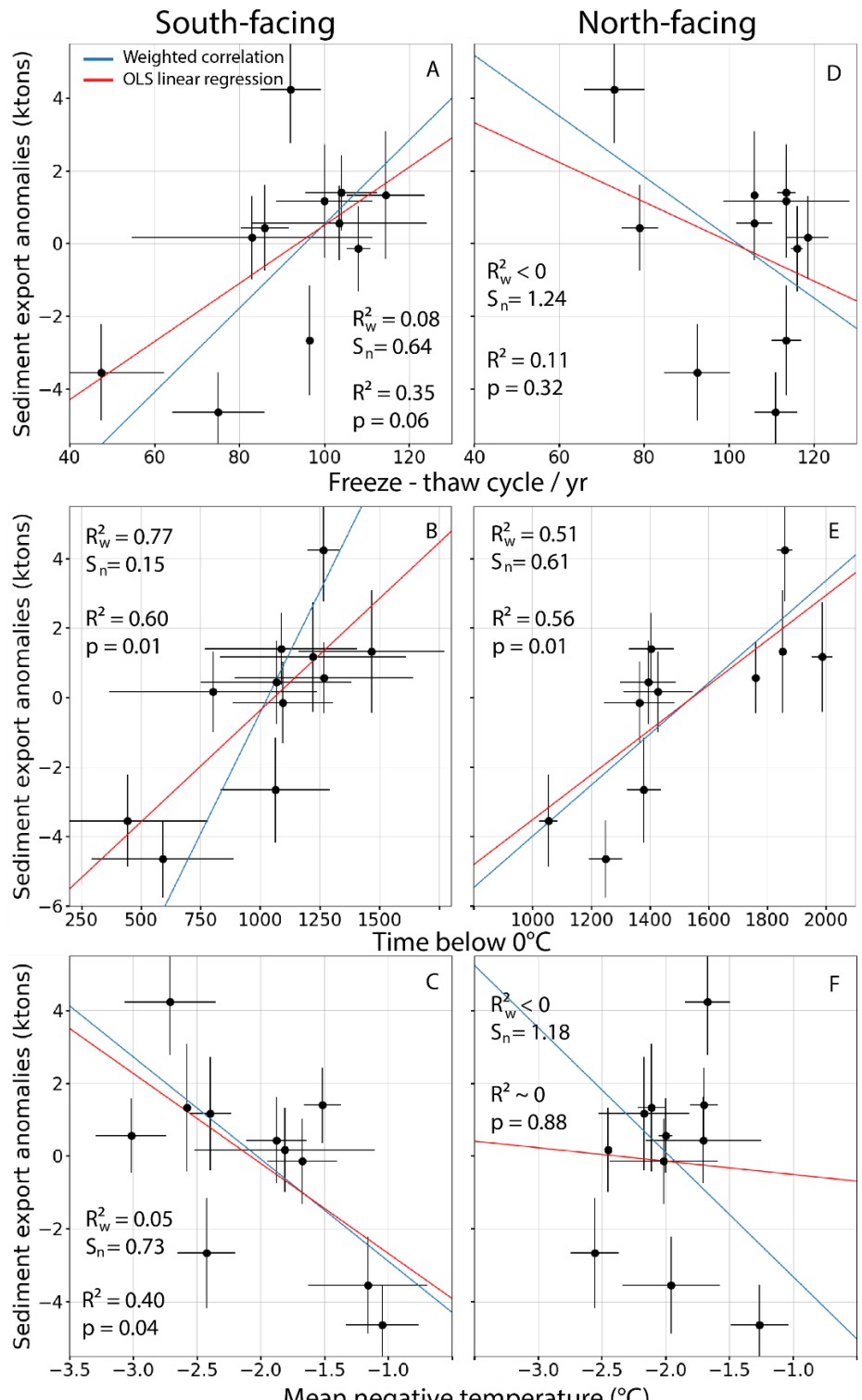

**Figure 9: Linear regression analysis between sediment-export anomalies and (A, D) number of freeze-thaw cycles per year; (B, E) time spent below 0 °C; and (C, F) mean negative temperature. Left column shows regressions for south-facing slope and right column for north-facing slope. Error bars and red and blue regression lines are as in Figure 8. Weighted determination coefficients $R^2_w$ and associated normalized goodness of fit indicators $S_n$ are indicated for each weighted regression. Standard $R^2$ and associated p-value indicate significance of the ordinary least-squares (unweighted) regressions.**


# Supplementary Material:

| Depht of probes | Full winter period used in the analysis | Aspects | Hill location | Reconstructed periods |
|---|---|---|---|---|
| -1 | 2006/2007, 2007/2008, 2008/2009, 2009/2010, 2011/2012, 2012/2013, 2013/2014, 2015/2016, 2016/2017, 2019/2020 | South-facing and North-facing | Uphill and Downhill | **2011/2012 SF uphill** : 24/12 6h30 to 31/12 23h50<br><br>**2011/2012 all**: 12/03 2h20 to 26/03 12h40<br><br>**2014-2015 SF**: 7/12 11h10 to 11/12 18h30<br><br>**2015/2016 all**:4/11 18h00 to 7/11 8h40, 21/11 00h00 to 14/12 15h20, 27/12 4h00 to 29/12 18h20, 01/019h10 to 5/01 23h50 |
| -24 | 2006/2007, 2007/2008, 2008/2009, 2009/2010, 2013/2014, 2014/2015, 2016/2017, 2019/2020 | South-facing only | Uphill (except for 2016/2017) and Downhill | No reconstruction possible |


**Supplementary Table 1: Information about temperature dataset available between 2006 and 2020.**


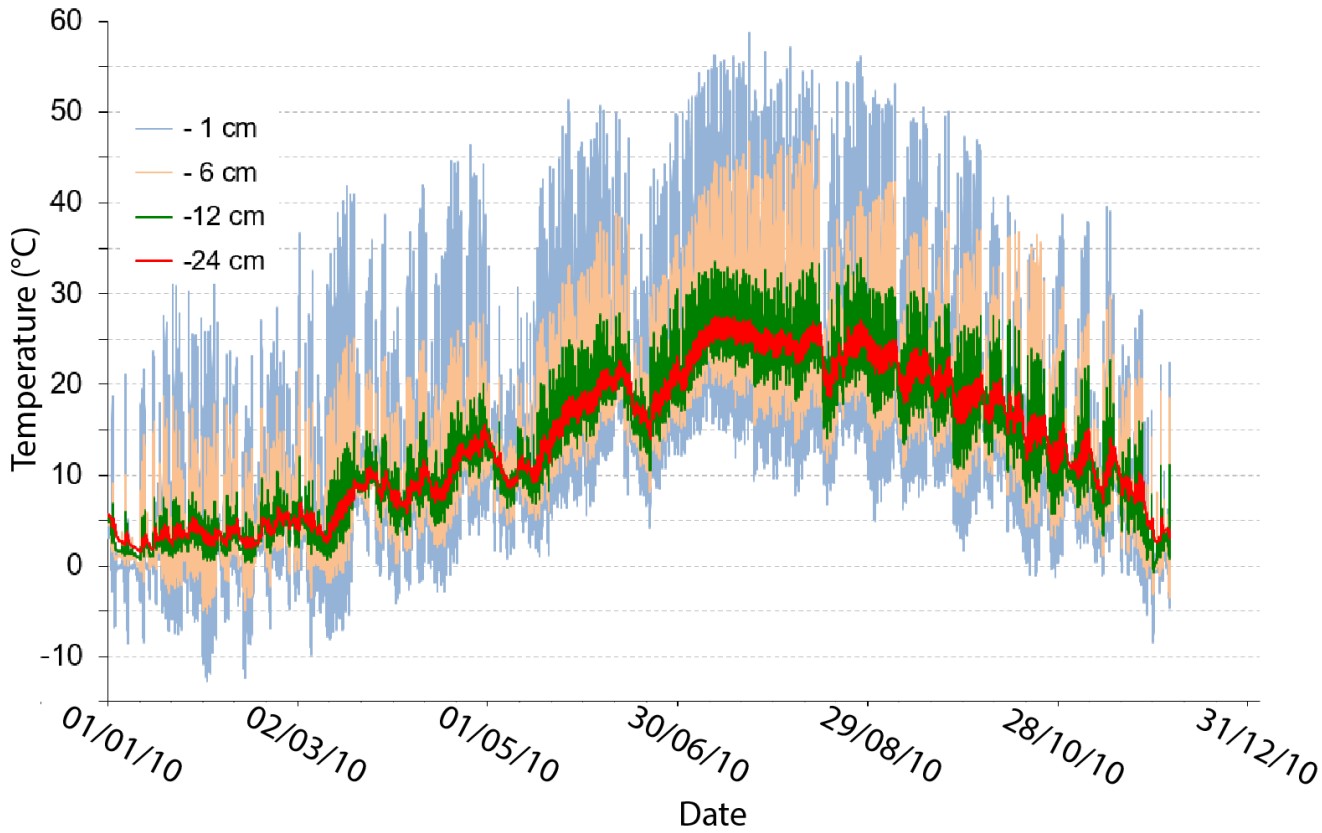

**Supplementary Figure 1: Annual time series of raw temperature data from four different depth probes.**


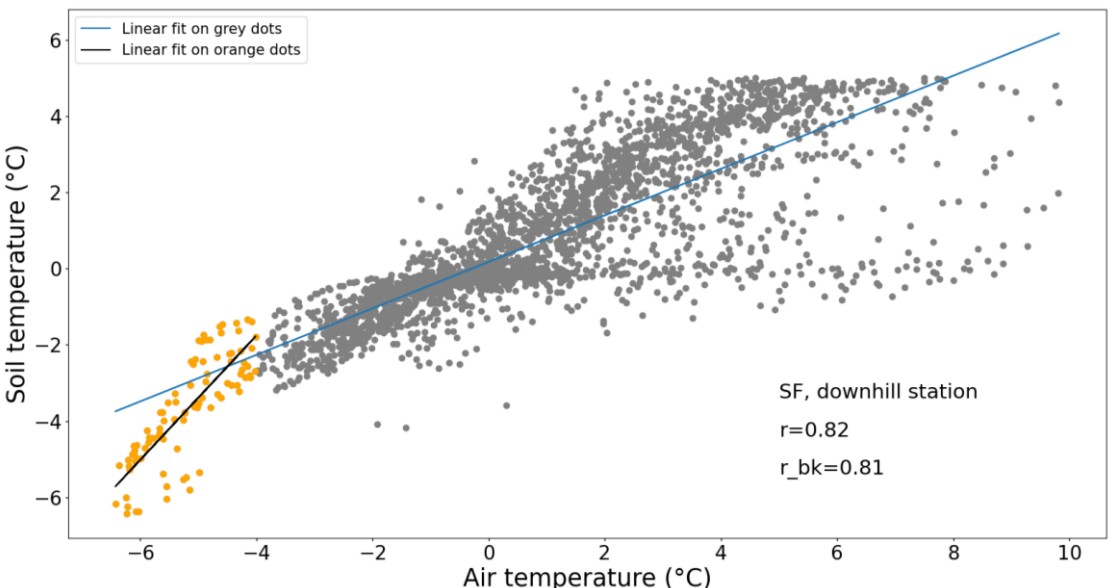

**Supplementary Figure 2: Correlation between soil and air temperature measurements on south-facing slope (SF) downhill station between 01/01/2014 and 31/03/2014. Since we are interested in frost weathering, temperatures were cut off at 5 °C for soil temperature and 10 °C for air temperature. We note a two-tier linear relationship between soil and air temperatures, with a steeper correlation for very low air temperatures. We therefore set a threshold for air temperature (-4°C) and regress the data above and below that threshold separately (grey / orange dots and blue / black lines, respectively; correlation coefficients r and r_bk for the higher- and lower-temperature regressions are indicated). The threshold was determined independently for every year where temperatures needed to be reconstructed. Dots aligned along 0 °C soil temperature record snow cover; Such snow-cover periods were avoided as much as possible when determining the linear regression parameters.**

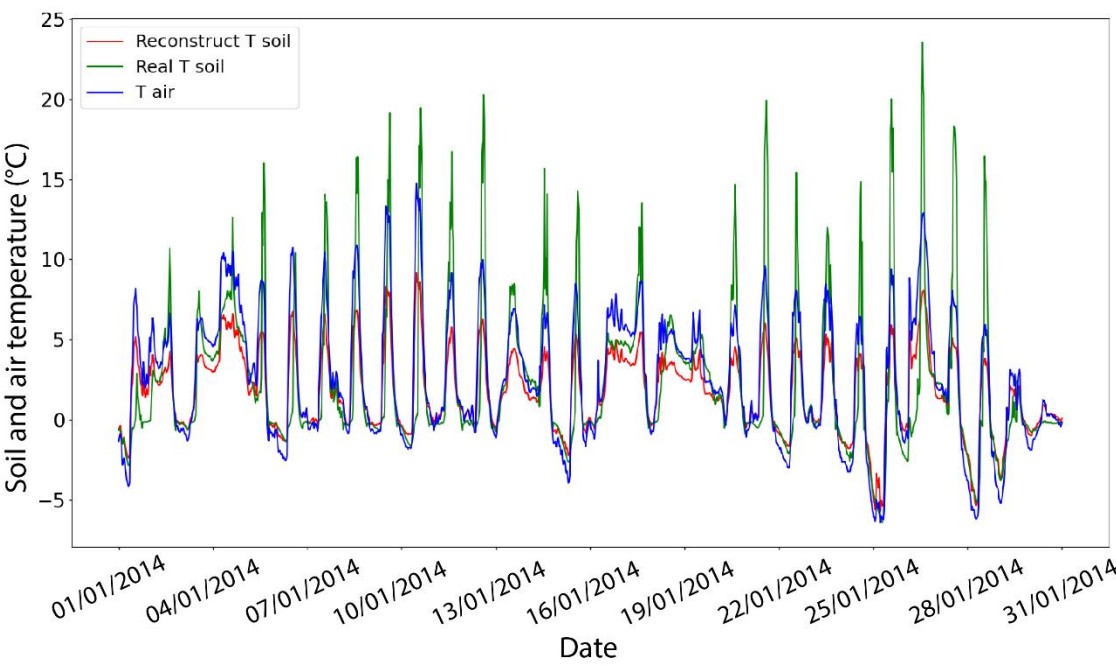

**Supplementary Figure 3: Comparison of measured soil temperatures (green line) and reconstructed temperatures (red line) for a complete temperature series (January 2014). Note that the model is only calibrated for soil temperatures <5 ℃ (Supplementary Figure 2). Air temperature is also shown (in blue).**



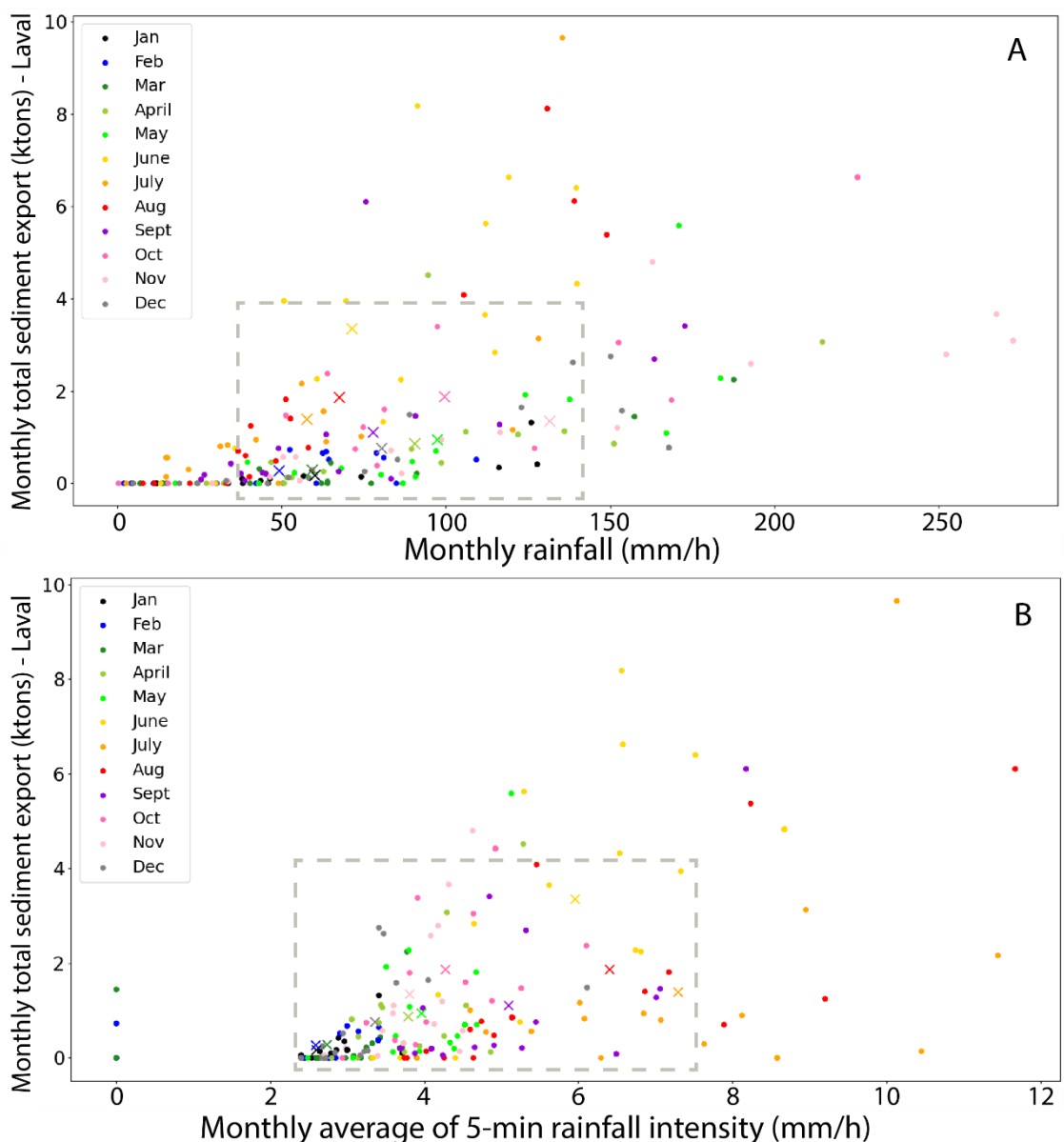

**Supplementary Figure 4: Monthly sediment export (ktons) from the Laval catchment versus (A) monthly rainfall between 2003 and 2020. (B) Monthly average of 5-min rainfall intensity (mm/h). Dots are individual values for each month and crosses are interannual monthly averages over the analysed period. Dashed boxes show limits of Figure 4A, B.**

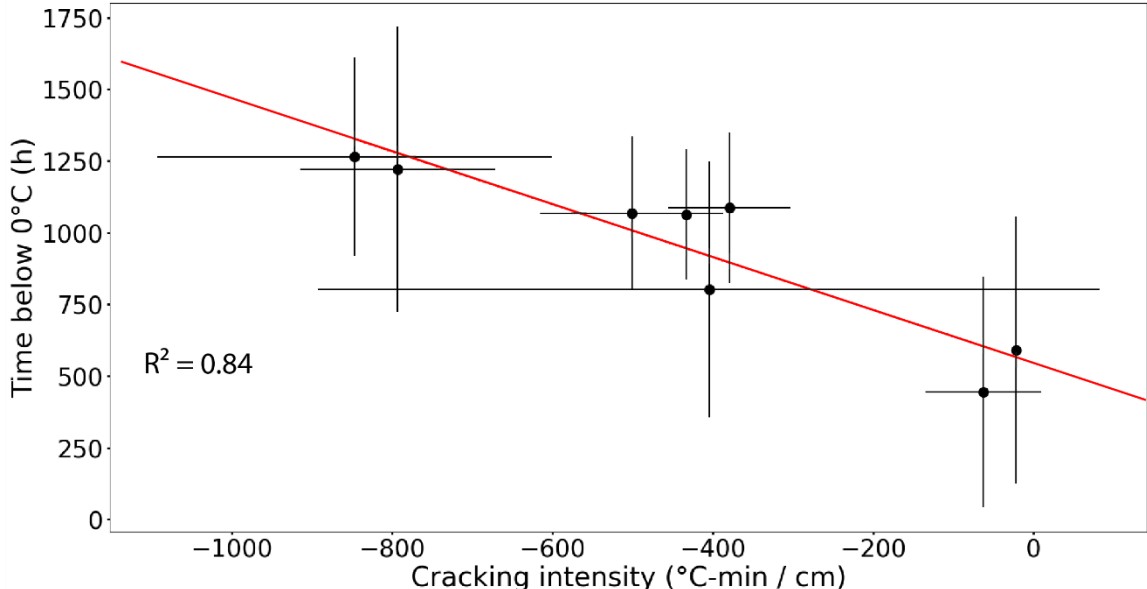

**Supplementary Figure 5: Regression analysis between frost-cracking intensity and the time below 0°C, the two temperature indicators that best predict the sediment export anomaly (Fig.7). Horizontal and vertical error bars refer to the difference between measurements at uphill and downhill locations (temperature measurements for 2017 and 2019 were only available for the downhill location; for these years the average difference for the other years was taken). Red line shows ordinary least-squares (OLS) linear regression giving a coefficient of determination $R^2 = 0.84$.**

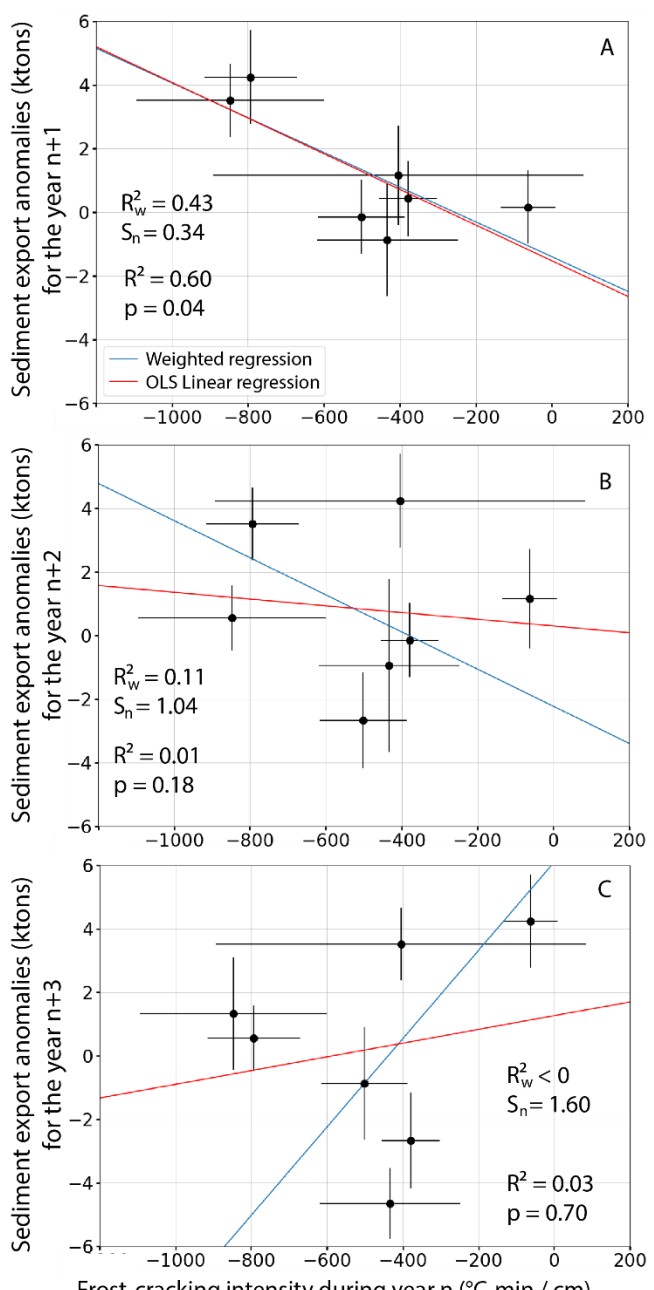

Supplementary Figure 6: Regression analysis between frost-cracking intensity on the south-facing slope for the years (n =) 2007 - 2010, 2014, 2015 and 2017 and the sediment-export anomalies of A) the year n+1, B) the year n+2, C) the year n+3 (See text for explanation). Horizontal error bars refer to the difference between measurements at uphill and downhill locations (temperature measurements for 2017 were only available for the downhill location, thus uncertainty on frost-cracking intensity was computed as the average of the uncertainties from the others years); vertical error bars are $\pm 2\sigma$ uncertainty in export anomaly. Red line shows ordinary least-squares (OLS) linear regression whereas blue line shows weighted linear regression following York et al. (2004). Weighted determination coefficients $R^2_w$ and associated normalized goodness-of-fit indicator $S_n = S / (n-2)$ are indicated for the weighted regression. Standard $R^2$ and associated p-value indicate significance of the ordinary least-squares (unweighted) regression.