# Peer review of "Sediment export in marly badland catchments modulated by frostcracking intensity, Draix-Bléone Critical Zone Observatory, SE France."

_Earth Surface Dynamics, 2021_

## Author Comment (AC1)

The manuscript esurf-2021-49 titled "Sediment export in marly badland catchments modulated by frost-cracking intensity, Draix-Bléone Critical Zone Observatory, SE France" presents a really interesting study based on a long-term database collected in a humid badland area located in the SE France. The manuscript presents interesting information and results, and I consider that it could be published after moderate changes. In my opinion, there are some major/moderate issues that should be checked before a new revision in Earth Surface Dynamics.

We thank the reviewer for their constructive comments, which we have addressed as explained below. In the following, the reviewer comments are indicated in black, our response in grey, and the proposed modifications to the manuscript in red in a grey italic paragraph.

1) In general, the manuscript presents new ideas and new data, analysis and results, but I will encourage the authors to highlight the novelty of your manuscript at the end of the Introduction section. In this paragraph, you should also improve the presentation of the general and specific objectives.

2) I think that it is really relevant the long-term dataset that you present in this manuscript. As you briefly mentioned in the manuscript, there are few sites well monitored and with such long-term dataset. In that sense, I think that in the introduction section, it should be remarked, and also in the discussion section (line 270). You can also include other areas/references where this kind of information is recorded (Tabernas in SE Spain, Vallcebre in NE Spain…).

We agree on the necessity to improve the presentation of the general and specific objectives of the study. In that sense, we suggest to modify the introduction as follows:

Line 84: *In this study, the general objective is to develop a similar approach at the catchment scale ($0.1$-$1$ km$^2$) in marly badlands. This objective is made attainable by the exceptional long-term dataset available in Draix-Bleone Observatory.* At the catchment scale, ...

Moreover, to highlight the challenge of obtaining such a long-term dataset, we propose the following modification :

Line 272: The Draix-Bléone CZO provides one of the few localities worldwide where such records exist. *Similar datasets have been collected in the Araguas and Vallcebre basins of (Northern Spain but cover significantly shorter periods ($6$-$18$ months; Regüés et al, 1995; Regüés and Nadal-Romero, 2013).*

3) In the study area, and throughout the manuscript (also in Figure 1) you present two experimental catchments (Laval and Moulin), but in most of the analysis (for example Figures 3 and 4) you only present data and information about the Laval catchment. This issue should be corrected or clarified in the manuscript.

Both the Laval and Moulin catchments are part of the Draix-Bléone observatory and are slightly differently instrumented. The soil temperature dataset that we used is recorded in the Moulin catchment and the climatological station is placed at the outlet of the Laval catchment, while sediment-export data is collected for both catchments. For this reason, it was necessary to present these two experimental catchments in the introduction (Fig. 1). Additionally, because of the availability of two separate datasets, it was possible to analyse the seasonal sediment dynamics in response to climatic events at different scales (Fig. 5). This approach allows us to conclude that these two catchments have the same general dynamics. We chose to work only

on the Laval sediment-export data for the correlation with temperature indicators for different reasons:

- Because of its larger size (0.86 km²), the Laval catchment is more representative than the Moulin catchment
- The vegetation cover is more limited on the Laval catchment(32%) compared to the Moulin catchment (46%) and, thus, a larger surface of the Laval catchment corresponds to the environment in which the temperature probes operate (bare marl bedrock ).
- The sediment dynamics are in any case similar for both catchments
- The climatic and soil temperature data is representative of both catchments, since they are very close to each other and share the same lithology
- The uncertainty is smaller for the sediment export form the Laval than for the Moulin 1) because the absolute values are higher and 2) because there are fewer floods where the suspended load data needs to be reconstructed

To clarify our approach, we suggest to add the following paragraph at the end of Section 3.1:

*The soil temperature data has been recorded in the Moulin catchment but is regarded representative also of the adjacent Laval catchment that shares the same lithology. Sediment-yield data is available for both the Moulin and Laval stations (located 100 meters apart at the outlets of both catchments; Fig. 1C) and our analysis of hysteresis cycles shows that both catchments have the same sediment dynamics (see Section 4.1, Figs. 4, 5). Because of its reduced vegetation cover and larger area (0.86 km$^{2),}$ we consider that the Laval catchment is more representative for the analysis of the relationship between sediment yield and temperature indicators (Figs. 3, 4 and 6 to 9) than the much smaller Moulin catchment (0.089 km$^2$). Moreover, the larger sediment export values from the Laval catchment are associated with smaller relative uncertainties. For these reasons, we focus our attention on the sediment-export data from the Laval catchment for our analysis.*

4) Line 115 and also check along the manuscript. You present the results about sediment yield in different units (tonnes, also erosion rates in mm/yr), I think that these results should be presented in specific values (tonnes per Km2 or ha), as the catchment sizes vary. Please, homogenize this information.

To take this comment into account, we suggest to modify the text (lines 115-118) as follows:

*The Draix catchments record some of the highest specific sediment yields observed worldwide: average annual sediment yields for the Laval and Moulin catchments are around 12,000 and 570 tonnes, **equating to specific sediment yields of around 14,000 and 5,700 tonnes/km$^2$/y** , respectively. Considering only the unvegetated parts of the catchments as contributing to the sediment yield **and the measured sediment density of 1700 kg/m$^3$**, the average erosion rate is around 8 mm/yr for both catchments (Mathys, 2006).*

5) It is also not clear how do you record the data in the field (around line 129). You explain that you record data during events but, do you have a continuous temporal record? It is true that the maximum sediment export values are recorded during flood events, but it would be necessary to understand how have you measured the data during all the study period. In that sense, it is always an interesting information to indicate which % of the sediment have been exported during the flood events, or even

which % of the sediment have been exported during the maximum events… in that
sense the high variability presented in Figure 3 could be better understood.

6) Information provided in lines 128-130 is already a result. In that sense, I suggest two
options: (i) to remove from this section and move to the results section; (ii) or to add
some reference where this information is already presented.

We do not have continuous temporal data on sediment export. The sediment trap stores bedload
continuously but topographic surveys are performed after each major flood event. Suspended-
sediment concentration is measured by turbidimeters only during flood events. However,
sediment export both as suspended load and as bedload is considered negligible during normal
flow in the study area. To support this hypothesis, we computed the annual sediment export
from the discharge and the concentration dataset between 2014 and 2018. Making the
hypothesis of an inter-event concentration between 0.1 g/L and 1 g/L (which is already a very
high value for low-flow), we find that only 0.12% to 1.2% of the total annual sediment export
is missing by neglecting the inter-event concentration as we did in the manuscript .

We therefore consider that the total sediment export is equivalent to the total sediment export
of the flood events reasonable. We added this information in the manuscript as follows:

*Line 126: Bedload volumes are measured after each flood by topographic surveys of a sediment
trap located upstream of the station. Bedload volume is then converted into mass using a density
of 1700 kg/m$^3$, constrained by measurements in the sediment trap (Mathys, 2006). **The raw data
we use is therefore a series of event-scale sediment yield. An analysis of inter-event sediment
export shows that flood export represents more than 99% of the total annual sediment export.
Thus,** sediment export both as suspended load and as bedload is considered negligible during
low flow and we define the total sediment export as the monthly or yearly sum of the suspended
load and bedload contributions **during floods**. For a few flood events, the suspended-load data
is missing. In such cases, we reconstructed the event-scale suspended sediment yield based on
the average proportions of suspended load and bedload, computed from multiple complete
years of total load records.*

7) Temperature depths and temporal range. Lines 134-140. I think you should better
justify the selection of the different depths why (1, 6, 12 and 24). Do you have any
reference to add to justify this selection? Or previous knowledge about it in the study
area? I'm also surprise about the time period selection, as you limited the information
from mid-October to the end of March. I understand your decision and your
justification, but it is true that in humid badland mountain areas there can be low
temperatures (below 0ºC) also in April or beginning of October.

Temperature probes were implanted in 2005 and the depths were selected to span the depth of
the loose regolith (Maquaire (2002)).

In order to justify our choice of time-span, we have now computed two indicators for the spring-
summer-autumn period (April 1$^{st}$ to October 17$^{th}$): the time spent below 0°C, and the time spent
in the frost cracking window ( -3° - -8°C).We show that these are both negligible with respect
to the values for these parameters during winter. To take these comments into account we
suggest to modify the text as follows:

*Line 138: At each location, four probes are available to measure soil temperature at depths of 1, 6, 12 and 24 cm, respectively,* **which span the ranges of depths that have been reported for the weathered regolith in this area (Maquaire, 2002).**

*Line 139: As our interest is on frost weathering, we specifically analysed soil temperatures during the winter season, from October 18th to March 31st. This period was chosen because negative soil temperatures are almost absent outside these dates.* **During the periods between April 1st and October 17th, we found that the time spent below 0 °C was on average less than 0.4% of the total time in a year and represents less than 4% of the time spent below 0 °C during the winter. The time spent in the frost-cracking window outside of the analysed winter period was null for most of the study years. We chose to start the winter season on October 18th because some yearly series miss temperature data for early October.**

8) Lines 144-152. More information about the reconstruction of the temperature data should be provided (even as supplementary material). Maybe, more info about this process should be included adding a plot with the calibration and reconstruction

To add more information about the reconstruction of the temperature data, we added two figures to the supplementary material and modified the text as follows in order to quantify the number of days that were reconstructed over the analysed period (winter period of each year):

[Figure]

*Supplementary Figure 2: Correlation between soil and air temperature measurements* on *south-facing slope (SF) downhill station between 01/01/2014 and 31/03/2014. Since we are interested in frost weathering, temperatures were cut off at 5 °C for soil temperature and 10 °C for air temperature. We note a two-tier linear relationship between soil and air temperatures, with a steeper correlation for very low air temperatures (r_bk = 0.81). We therefore set a threshold for air temperature (-4°C) and regress the data above and below that threshold separately (grey / orange dots and blue (r= 0.82) / black lines, respectively). The threshold was determined independently for every year where temperatures needed to be reconstructed. The red ellipse show dots aligned along 0 °C soil temperature and record snow cover . Such snow-cover periods were avoided as much as possible when determining the linear regression parameters.*

[Figure]

*Supplementary Figure 3: Comparison of measured soil temperatures (green line) and reconstructed temperatures (red line) for a complete temperature series (January 2014). Note that the model is only calibrated for soil temperatures <5 °C (Supplementary Figure 1). Air temperature is also shown in blue.*

Modification in the text:

*Line 147: When the missing period is shorter (seasons 2011/2012, 2014/2015, 2015/2016), we searched for a relation between soil temperature and air temperature (recorded at the closest weather station) in order to reconstruct soil temperature during the missing time interval* **(Suppl. Figures 2 and 3). Over the eleven winter seasons used to calculate our temperature indicators, we reconstructed 56 days, which represents around 3% of the total winter temperature dataset (see Supplementary Table 1 for details).**

9) I think that there is also some confusion about the period you used in the analysis. Some times you indicate 2005-2019 (line 160) but then you indicate 2003-2019 (210). Maybe, as suggestion, it would be nice to include a table (maybe in the Supplementary material) with information about the variables that you have used, where (south-facing or north-facing, high-low slope), and the period that have been used for the analysis. In that sense, it would be really clear.

We used two different time periods for our analyses:

- 2003 to 2019 for the monthly sediment export analysis (4.1, 5.2: Figs. 3 - 5)
- 2005 to 2019 for the temperature indicator analysis (4.2, 5.3 : Figs. 6 - 9)

Because these two results are not directly correlated we used the longer time range available for the sediment-export data to analyse the hysteresis cycle. However, when comparing annual

sediment export with the temperature indicators (Fig. 6) we only used the data in the timespan 2005-2019 in order to be consistent with the time range of the temperature datasets.

To reduce the confusion concerning these different time ranges, we added some information at the beginning of Section 3.2.1 part (line155) and mentioned our work on monthly sediment export:

*First, we compared monthly sediment export and monthly rainfall to understand the seasonal dynamics of sediment transport in the Laval and Moulin catchments (Figs. 3 - 5). The analysis was performed for the period 2003 - 2019, during which sediment export was precisely recorded.*

10) In that sense, it is also not clear which information have been obtained and analyzed in north- and south-facing slopes. It is really important, as the literature already have shown that there are significant differences between both expositions. So, if you only calculated some indices for south-facing slopes, some part of the history is missing and it is a pity. Please, think about it because in lines 250-255, you indicate that better correlations are obtained in south-facing slopes, but I think it is not totally true, as one of the variables have not been measured/calculated in the north-facing slope.

We are not sure how we can make this more clear, as we already indicate very clearly in lines 175, 247 and 250-255 what temperature indicators were measured on the north and south-facing slopes, respectively. We acknowledge that it is unfortunate that the deeper temperature probes malfunctioned on the norht-facing slope, disallowing us to calculate FCI on this slope

To take in account your concerns, we slightly modified the text (line 252) and added a recapitulative table on the temperature dataset in the supplementary material.

**Line 250 -255**: *On the north-facing slope, in contrast, the only significant correlation between temperature indicators occurs between the number of freeze-thaw cycles / year and mean negative temperature (R= -0.62).* *On south-facing slope, the correlation between indicators and sediment-export anomalies show the strongest correlation (R= 0.87) for the frost-cracking intensity indicator but time below 0 °C indicator is also significantly correlated (R= 0.71). On the north-facing slope, the only significant correlation occurs between sediment-export anomaly and time below 0 °C (R= 0.74), but note that frost-cracking intensity was not calculated on the north-facing slope* *(Supplementary Table 1).*

*Supplementary Table 1: Information about temperature dataset available between 2005 and 2019*

| Depth of probes | Full winter period used in the analysis | Aspects | Hill location | Reconstructed periods |
|---|---|---|---|---|
| -1 | 2006/2007, 2007/2008, 2008/2009, 2009/2010, 2011/2012, 2012/2013, 2013/2014, | South-facing North-facing | Uphill Downhill | **2011/2012 SF uphill** : 24/12 6h30 to 31/12 23h50 **2011/2012 all**: 12/03 2h20 to 26/03 12h40 **2014-2015 SF**: 7/12 11h10 to 11/12 18h30 |

| | | | |
|---|---|---|---|
| | 2015/2016, 2016/2017, 2019/2020 | | | **2015/2016 all**:4/11 18h00 to 7/11 8h40, 21/11 00h00 to 14/12 15h20, 27/12 4h00 to 29/12 18h20, 01/019h10 to 5/01 23h50 |
| -24 | 2006/2007, 2007/2008, 2008/2009, 2009/2010, 2013/2014, 2014/2015, 2016/2017, 2019/2020 | South-facing only | Uphill (except for 2016/2017)  Downhill | No reconstruction possible |

**11)** I suggest to include the complete temperature database (or at least the period you consider) in the manuscript and not only 2-3 months (Figure 2). I think it is really important to show the temperature pattern in this area, at different depths, once you have already measured this variable. And also the information about the different temperature variables that you have measured (maybe boxplot or similar plots). I'm also really surprise about the high temperatures you recorded in the winter months (January temperature higher than 30º). In Figure 2, you should indicate if these data are average daily temperatures.

We chose to show only a short period for the temperature database in order to more clearly identify the different patterns of each probe. We thought that the annual cyclic variations of temperatures were less relevant for this study. However, in response to this comment, we have added a graph showing a complete annual temperature series to the Supplementary material, allowing the reader to visualise the lower-frequency trend of temperatures during an entire year.

Moreover, we added to the legend of Figure 2 that data are recorded every 10 min. The high peak temperatures of the -1 cm probe (even during winter) are due to radiative heating of the heat-absorbing black marls during sunny periods.

*Figure 1: Example of raw temperature series (1 measurement every 10 min) (A) Typical soil-temperature series recorded with four probes at different depths (from south-facing uphill location). (B) Example of soil-temperature series (from south-facing downhill location) biased because of climatic conditions (snow cover), buried or loosened probes. A full year of temperature measurements is shown in Supplementary Figure 1. High temperature values are observed at -1cm even in winter when black marl heats up in sunny periods.*

[Figure]

*Supplementary Figure 1: Annual time series of raw temperature data from four different depth probes.*

12) In the analysis presented in Figure 6, and also explained and discussed in the text, it is not clear what is the meaning of rainfall above 50 mm. Please could you define it? Rainfall events, flood events with rainfall > 50 mm. In that sense, Figure 6 X axis is the number of rainfalls above 50 mm/h. Please specify it in the figure caption and clarify in the manuscript.

To clarify Figure 6, we will modify the x-axis title, the caption and text as follow:

*x-axis title: Cumulative rainfall (mm) above a threshold*

*Caption Figure 6: Linear correlation between annual total sediment export from the Laval catchment and **the cumulative rainfall above an instantaneous intensity threshold of 50 mm/h** for the years 2005 to 2019 (blue dots). Regression line is in black; grey shaded area shows 95% confidence interval. Most outliers occur for low cumulative rainfall above the threshold (< 40 mm). (B) Coefficient of determination (R²) between annual sediment export and cumulative annual rainfall above threshold for different intensity thresholds. Optimum correlations are found for intensity-threshold values between 50 and 55 mm/h.*

We also made this variable more explicit in the text line 228:

*Based on annual data records since 2005 and previous work (see Methods section), we established a correlation between **the cumulative rainfall above an instantaneous intensity threshold and sediment export (Fig. 6).***

13) Discussion sections 5.2 and 5.3. Once I have read the discussion 5.2, I really miss some information about geomorphological dynamics in humid badland areas and explanations about weathering processes, including some references with similar studies in humid badland areas worldwide. It is true, that then once I started reading section 5.3 I partially found this information. I suggest to link both sections and organize the ideas presented in both previous sections.

- o In that sense, some significant references that I missed in the manuscript and that could be checked are:
  - Clarke and Rendell, 2006. Process-form relationships in Southern Italian badlands: erosion rates and implications for landform evolution. ESPL 31.
  - Clarke and Rendell, 2010. Climate-driven decrease in erosion in extant Mediterranean badlands. ESPL 35.
  - Nadal-Romero and Regüés, 2010. Geomorphological dynamics of subhumid mountain Badland areas- weathering, hydrological and suspended sediment transport processes: A case study in the Araguás catchment (Central Pyrenees) and implications for altered hydroclimatic regimes. Progress in Physical Geography 34.
  - Gallart et al., 2013. Thirty years of studies on badlands, from physical to vegetational approaches. A succinct review. Catena 106.
  - Gallart et al., 2013. Short- and long-term studies of sediment dynamics in a small humid mountain Mediterranean basin with badlands. Geomorphology 196.
  - Bollati et al., 2019. Alpine gullies system evolution: erosion drivers and control factors. Two examples from the western Italian Alps. Geomorphology 327.
  - Llena et al., 2020. Geomorphic process signatures reshaping sub-humid Mediterranean Badlands: 2. Application to 5-year dataset. ESPL 45.
  - Llena et al., 2021. Do Badlands (always) control sediment yield? Evidence from a small intermittent catchment. Catena 198.

We feel that paragraphs 5.2 and 5.3 treat quite separate subjects and, as such, we prefer to keep them separate. However, upon rereading paragraph 5.2, we agree it can be sharpened by more appropriate comparisons with other similar study sites. Thus, we have studied the references provided by the reviewer and incorporated these in a revised version of paragraph 5.2.

Line 301: *Numerous studieshave reported annual hysteresis cycles between rainfall or discharge on one hand, and sediment export on the other; these studies, have commonly focused on large catchments, e.g. in the Andes or Himalaya (e.g., Andermann et al., 2012; Armijos et al., 2013; Tolorza et al., 2014; Li et al., 2021). For such large catchments, the annual hysteresis cycle is explained by the role of subsurface water storage (Andermann et al., 2012), dilution effects (Armijos at al., 2013) or variations in the contributive erosive area (Li et al., 2021). Hysteresis cycles for smaller catchments have generally been analyzed at the event scale, and have been interpreted in terms of the proximity of sediment sources and the spatio-temporal heterogeneity of rainfall (Klein, 1984; Buendia et al., 2016).* More directly comparable to our results, several studies have been carried out in small (< 15 km²) Mediterranean badland catchments where climate can vary between arid to humid conditions. Llena et al. (2021) reported a seasonal sediment dynamic with lags between sediment production and sediment yield and highlighted the role of the channel network in the sediment transfer. Several catchments in Northern Spain that are very similar in size and lithology to our study site have

been well studied and suspended sediment transport processes have been reconstructed at the event (Soler et al., 2008; Nadal-Romero et al., 2008) and seasonal scale (Nadal-Romero and Regüés, 2010). Counter-clockwise and clockwise hysteresis loops in these catchments are associated to dry and wet seasons, respectively, and are inferred to be driven by infiltration and saturation processes on hillslopes.

The annual hysteresis cycle between rainfall and total sediment export observed in the Draix-Bléone CZO (Fig. 4A) presents two loops with successively anti-clockwise and clockwise patterns, reflecting the rapid seasonal changes in erosion regime in these badlands

---

## Author Comment (AC2)

Dear Editor, dear authors,

The paper "Sediment export in marly badlands catchments modulated by frost-cracking intensity, Draix-Bléone Critical Zone Observatory, SE France" presents a very valuable data set that will be of the interest of the ESurf readership. Its conclusions regarding frost-cracking models and potential alternative proxies are relevant for the community and a timely publication. It is overall well-written and I enjoyed reading it. However, I recommend to the Editor that the manuscript is returned to the authors for Minor Revisions before publication. The suggestions I make below are mostly towards improving the clarity of the aim, hypotheses, and writing in some parts, and including a few additional figures.

We thank the reviewer for their constructive comments on the manuscript. In the following, the reviewer comments are indicated in black, our response in grey, excepts from the manuscript in grey italic, and the proposed modifications to the manuscript in red italic. Modifications in response to comments from reviewer #1 are indicated in green.

1) My main concern is that the aim and hypotheses of the study are not clear in the abstract and introduction, and hypotheses of this study are hard to untangle from the conclusions of previous studies. For example, in lines 11-14 of the abstract, it says "rainfall variability does not fully explain (…) sediment export (…) suggesting that sediment production may modulate (…) sediment export". **My first thought was that at this stage of the abstract, why ruling out the influence of other potential processes such as sediment storage**? Later, in lines 84-85, it is clear that sediment storage is not relevant due to the very small scale of the studied catchments. But because this is not mentioned in the abstract, as it is written now, it's quite puzzling to the reader to **see only one hypothesis favoured.**

2) The abstract sound like the aim is to explore what controls the sediment export anomalies, but the first paragraphs of the introduction show **that you have already, before doing any research, a pretty good idea of what controls those anomalies from previous published research** (e.g. "Rovera and Robert (…) noted the marls' sensitivity to frost weathering…" (line 50); then line 55 "(…) inferred that these catchments was mainly dependent on the number of freeze-thaw cycles occurring during the year" lines 55-61 " Bechet et al. (…) inferred a yearly cycle between transport-limited conditions in spring to supply-limited conditions in autumn"). After the first page of the introduction, **I get the impression that this is a very-well studied field site and I don't understand what the aim of this current study is.** When in line 86 it says "we hypothesize that frost-weathering processes can modulate sediment yield by controlling regolith production on hisllslopes", this sounds too similar to the conclusions of previous studies and I don't understand why this is a hypothesis that needs testing if it has already been shown to be true. **The goals of the study (lines 90-93) also need a bit more differentiation from what's already known in the study area so that the relevance of this study can come across more clearly.**

To clarify the context of the study and the current knowledge gaps, the abstract has been modified in order to better outline the main outstanding problems in these badland areas, exposed in the introduction, and the (more qualitative) outcomes of previous studies that motivate our research questions, exposed at the end of the introduction (L77-80; 84-93).

Line 9: *Long data records (30 consecutive years for sediment yields) collected in the sparsely vegetated, steep and small marly badland catchments of the Draix-Bléone Critical Zone*

*Observatory (CZO), SE France, allow analysing potential climatic controls on long-term regolith dynamics and sediment export. Although widely accepted as a first-order control, rainfall variability does not fully explain the observed inter-annual variability in sediment export. Previous studies in this area have suggested that frost-weathering processes could drive regolith production and potentially modulate the observed pattern of sediment export. Here, we address this question quantitatively, by defining sediment-export anomalies as the residuals from a predictive model with annual rainfall intensity above a threshold as the control. We then use continuous soil-temperature data, recorded at different locations over multiple years to highlight the role of different frost weathering processes (i.e., ice segregation versus volumetric expansion) in regolith production.*

Whereas the reviewer mentions our hypothesis L86, they ignore the previous sentence where we explained that one of the main advances of our work is the catchment-scale analysis, as opposed to the plot- or hillslope-scale analyses of previous studies. However, we note that both reviewers commented on the lack of clarity of the objectives of the study; we took these comments into account to clarify the general and specific objectives as follows:

*Line 84: In this study, the general objective is to develop a similar approach at the catchment scale (0.1-1 $km^2$) in marly badlands, taking advantage of the exceptional long-term dataset available for the Draix-Bléone Observatory. At the catchment scale, sediment export is primarily driven by rainfall, particularly during high-intensity events (Mathys et al., 2003), but we hypothesize that frost-weathering processes can modulate sediment yield, even at this scale, by controlling regolith production on hillslopes. Coupled to this first hypothesis, this study aims to highlight and quantify the main frost-weathering process in a setting of humid climate and soft lithology, by using high-resolution soil-temperature measurements.*

3) Likewise, if the aim of the paper is to explore the controlling factors on sediment production/export, when I read that the summer surface temperature show a variability of 40-50 degrees Celsius (lines 239-237), I was surprised that, considering the data set available, solar-induced thermal stresses and their effect on physical weathering have not been considered as another potential variable influencing catchment sediment export (e.g. see Missy Eppes' papers).

We thank the reviewer for this interesting comment. However, several arguments can be put forward to justify our choice of focussing on frost-weathering processes rather than solar-induced thermal stresses:

- The Eppes et al. (2016) experiments were conducted on a granite boulder, a very different lithology from the marls we deal with (in terms of texture, structure, physical properties, etc); we therefore feel that these results cannot be simply transposed to our study context.
- Saprolite (widely fractured marls) contains significantly more liquid water than granite cracks, and we infer that frost weathering is therefore much more efficient in these lithologies.
- Bechet et al. (2016) demonstrated sediment accumulation on hillslopes during the winter, pointing toward winter soil production.
- Consistent with the previous point, the hysteresis cycle of monthly sediment export (Figure 4) shows that a supply-limited regime is installed by the end of the summer

period. This observation is in contradiction with potential sediment production by solar-induced thermal stresses during the summer (i.e. the, season when the highest temperatures are recorded). This argument will be add in the text as follows:

Line 340: *Together with the evidence for a transition from transport-limited to supply-limited conditions during the year discussed above, we interpret these results as indicating that frost-weathering processes modulate sediment export from the catchments by exerting a strong control on the production of mobilizable sediment.* ***In particular, the lack of sediment supply during summer months inferred in the previous section argues against significant sediment production by solar-induced thermal stresses (e.g., Eppes et al., 2016), despite high daytime surface temperatures and large temperature variations in the marls during summer (Suppl. Fig. 1).***

Testing the volumetric expansion and ice-segregation frost-cracking models in "temperate/humid climate and soft lithologies" (Lines 77-79) is perhaps the most novel aim of this manuscript, and one that seems to adjust better to its content. However, this does not come across clearly in the abstract and introduction. I suggest rephrasing to put more emphasis on this, and carefully rewriting the current aim and hypotheses to more clearly differentiate them from the conclusions of previous studies.

This comment was taken into account; see our responses to points 1 and 2 above.

   4) In section 2, a bit more context on the catchments geomorphology (range and mean hillslope angles?) and dynamics (are there any frequent small landslides? Are the streams ephemeral?) would also be useful.

Both in the Moulin and Laval catchment, convex forms are common and hillslopes are steep (mean slopes: Laval:0.58; Moulin: 0.40). Hillslopes become steeper with increasing elevation and thus range between …..We suggest to modify the text as follows:

Line 104: *The rainfall regime varies across seasons, with high-intensity rainfall events during spring/summer and lower-intensity but longer rainfall events in autumn.* ***Only the main streams (Laval and Moulin) are permanent, although the Moulin shows very small discharge in summer; all tributary gullies are ephemeral.***

Line 113: *These black marls are susceptible to strong erosion and develop steep badland slopes* ***(mean hillslope angles of 0.58 for the Laval and 0.40 for the Moulin catchment),*** *with high drainage density and deeply incised gullies characterizing the catchment morphologies (Fig. 1C, D).* ***Sediment transport occurs through gravitational processes on hillslopes, minor landslides (<1 m³), micro debrisflows in the upper network, and bedload and suspended load in the main network.***

   5) In the methods, the timing of data set acquisition is a bit unclear, see my line-by-line comments below.

As we received similar comments from reviewer #1, we modified the text (line 250) and added a supplementary table (Supplementary Table 1) to clarify the timing of dataset acquisition.

   6) Figures. 3 and 4 show monthly total rainfall, but figure 6 focuses on rainfall intensity. It would be useful to show for reference a box plot of the distribution of rainfall intensity

across the months (e.g. do October and November have also higher rainfall intensity, or just higher total rainfall because it rains more days?). It would also be useful to see how panel B of Fig. 4 looks like when average monthly rainfall intensity is plotted rather than total monthly rainfall. This would also help better visualizing and following the discussion of section 5.2.

We thank the reviewer for this very interesting comment. As suggested, we have computed and plotted the monthly-averaged rainfall intensities. Because the calculation of the inter-annual monthly average of the rainfall instantaneous intensity was not relevant (variable time step), we work with rainfall intensity at 5-min constant time-step in order to compute monthly average rainfall intensity. The following figures, which show the results of this computation, will be modified / added in the manuscript and the previous Figure 4 will be moved to the supplementary material. The boxplot analysis highlights the seasonal variations of the rainfall intensity with the highest values (late spring- early summer), associated to highest variability, twice higher than winter values. Autumn mean rainfall intensity is clearly smaller than summer intensity, which contrast with the monthly cumulative rainfall (Figure 3A et 4B).

Note that the rainfall intensity that we have used for these monthly averages is computed over 5 minute time-steps, contrary to the data used for Figure 6 that was based on instantaneous intensities (i.e. computed at the variable time step of each tipping event). This is mentioned in the captions of the figures.

[Figure]

*Figure 3: Boxplots of (A) monthly rainfall, B) Monthly average rainfall intensity computed at 5-minute time-steps and C) monthly total sediment export (i.e., bedload + suspended load) of the Laval catchment. White line shows median value and white dot indicates mean value. The first (Q1, 25%) and third (Q3 75%) percentiles are indicated by the box limits, whiskers show Q1 – 1.5\* IQR and Q3 +1.5\*IQR (inter-quartile range). Black dots are outlier values.*

[Figure]

*Figure 4: Hysteresis plot using interannual monthly average values from the Laval catchment between 2003 and 2020. (A) Monthly sediment export (ktons) versus monthly rainfall. Dashed line highlights the hysteresis cycle with two separate maxima: high sediment export and moderate rainfall in June versus high total rainfall and moderate sediment export in October/November (B) Monthly sediment export (ktons) versus monthly average rainfall intensity. Dashed line illustrate the hysteresis cycle with a maxima of sediment export in June preceding a stationary stat around 1.5 ktons of sediment export between July and November.*

In light of this observation, we suggest to modify and add details about these results in the text of the manuscript part 4.1 and 5.2 (see in the new version of the manuscript) and in part 3.2.1 as follows:

Line 155*: First, we compared monthly sediment export and monthly rainfall to understand the seasonal dynamics of sediment transport in the Laval and Moulin catchments (Figs. 3 - 5). The analysis was performed for the period 2003 - 2020, during which sediment export was the most precisely recorded.* **In this time interval, rainfall amount was summed for each month to obtain monthly rainfall. When gaps were present in the data (around 6% of the time), daily cumulative reconstitutions of these missing periods were possible thanks to the network of tipping-bucket**

*rain gauges installed around the catchment. Monthly averaged rainfall intensity was also computed by averaging non-zero values of 5-minute constant time-step rainfall intensities.*

7) Finally, showing a direct correlation of time spent below 0 degrees and frost-cracking intensity would add value to the paper and convey one of its most important outcomes more easily.

The direct correlation between the time spent below 0°C and the frost-cracking intensity is already illustrated by the correlation matrix (Figure 7; significant value of -0.92). We are not sure that adding a figure showing this correlation would add much but we suggest adding it as a supplementary figure, which we refer to in the discussion:

[Figure]

*Supplementary Figure 5: Regression analysis between frost-cracking intensity and the time below 0°C, the two temperature indicators that best predict the sediment export anomaly (Fig.7). Horizontal and vertical error bars refer to the difference between measurements at uphill and downhill locations (temperature measurements for 2017 and 2019 were only available for the downhill location; for these years the average difference for the other years was taken). Red line shows ordinary least-squares (OLS) linear regression giving a coefficient of determination R² = 0.84.*

Line 356 : *However, our study suggests that the time spent below 0°C, which correlates well with the frost-cracking intensity (Supplementary Figure 5), may be used as a simpler proxy to predict frost-weathering intensity.*

LINE-BY-LINE COMMENTS:

For every line-by-line comment we suggest directly a modification of the manuscript, if needed.

8) Lines 9 and 11: when saying "long data records" and "long-term" please add in parenthesis an actual timescale bracket. Otherwise it's a bit vague and different people understand different things by "long-term".
See the modification made in response to comments 1 and 2.

9) Line 43: "very large quantity of sediment" – again, this sounds a bit vague and open to interpretation, add some order of magnitude or range in parentheses

10) Lines 42-44: to follow this sentences better and to provide more context, it would be useful for the reader to know what's the magnitude and frequency of these floods typically

We added in this part quantitative value associated to a typical big flood (in the top 10 of the floods for bedload records since 1995) from spring 2014. For one flood event, the total sediment-yield (suspended load and bedload) was around 5960 tonnes, giving a total specific sediment–yield of 6930 tonnes/km². We suggest to add this quantitative information in the introduction part and give more context on magnitude and frequency of these floods in the section 2.

Line 42: *Because of the ample availability of sediment and the efficient network connectivity (Jantzi et al., 2017), floods in these catchments can transport a very large quantity of sediment (Delannoy and Rovéra, 1996).* **As an example, during one flood event on 17/06/2014, 6390 tonnes/km² were exported and the suspended sediment concentration reached 440g/L***).*

Line 115*: The Draix catchments record some of the highest specific sediment yields observed worldwide: average annual sediment yields for the Laval and Moulin catchments are around 12,000 and 570 tonnes,* *equating to specific sediment yields of around 14,000 and 5,700 t/km²/y,* *respectively.* **This sediment budget results from 22 floods per year on average (for the Laval), ranging between 13 and 45 floods / yr and associated with very heterogeneous sediment yields from 0 up to around 6500 t/km² per event (Smetanova et al, 2018)***. Considering only the unvegetated parts of the catchments as contributing to the sediment yield and the measured sediment density of 1.7 kg/m³, the average erosion rate is around 8 mm/yr for both catchments (Mathys, 2006).*

11) Line 51: "distinct evolution between S-facing and N-facing slopes" – distinct how? In the time, the magnitude of the response,….?

We cite Rovera and Robert (2005) in order to head the reader towards the details of the study and make our writing more fluent. We suggest to lightly modify the manuscript as follows:

Line 50: *Rovéra and Robert (2005) first investigated periglacial erosion processes in the Draix-Bléone CZO; they noted the marls' sensitivity to frost weathering,* **in particular to freeze-thaw cycles, and the resulting faster ablation** *on north facing than south-facing slopes*

12) Line 55: how many years of observations does this time-series contain?

Line 55*: Based on* **a two-year** *time series of* **twelve** *high-resolution digital elevation models from a 0.13 ha catchment in the Draix-Bléone CZO, Bechet et al (2016) showed…*

13) Line 62: what other similar sites? Please provide a couple of examples or delete this part of the sentence

'Other similar sites' refer to Spanish badlands mentioned in the previous paragraph.

Line 62: *Overall, existing observations from the Draix-Bléone CZO and similar sites* **(e.g., Regüés et al., 1995; Nadal-Romero et al., 2007)** *lead to consider frost weathering as a potentially important process controlling regolith production in marly Alpine badlands.*

14) Line 87: what is meant by "high-resolution" – give some more specific indication of the resolution.

More details on the -soil temperature series are given in part 3.1 (Line 134), thus we made a minor modification to the introduction.

Line 86: *To test our hypothesis, we used high-resolution soil-temperature measurements* **(every 10 min)** *from different locations and compared calculated temperature indicators,*

15) Line 89: what winter season and "following year"? Would be handy to know the exact year.

Our response is similar to the previous point: more details are provided in the third part of the manuscript but we add some detail here.

Line 88-89: *'...including the number of freeze-thaw cycles, the time spent below 0 °C, the mean negative temperature and the frost-cracking intensity, during a winter season to the sediment-export anomaly (i.e., the residual of sediment yield that cannot be explained by rainfall variability) in the following year* **(e.g., for the 2007-2008 winter season, we used the sediment export for the year 2008)**.

16) Line 109: is the mean elevation of one catchment 850 and for the other, 1250 (and if so, for which), or does the elevation of both catchments range from 850 to 1250 m of elevation? Please rephrase to make this clearer.

Line 109: *Elevations* **range from 850 to 1250 m a.s.l. for the Laval and from 850 to 925 m a.s.l. for the Moulin catchment.**

17) Line 114: and how much would the average erosion rate be if the whole catchment is considered? Most studies do not remove vegetated sectors when providing catchment-averaged erosion rate estimates.

We considered only the unvegetated part of the catchment for the erosion rate computation because it has been shown that the vegetated parts do not contribute to the sediment-yields. As a witness, the vegetated Brusquet catchment has a specific sediment yield about two orders of magnitude less than the Laval catchment sediment yield, or about 0.5% (Carriere et al, 2020). In that sense, we suggest to not modify the given erosion rate but we add details on this part as follows:

Line 117: **It has been shown that erosion is strongly focused in the unvegetated parts of the catchments; the specific sediment yield of the adjacent vegetated Brusquet catchment is two orders of magnitude smaller than that of the Laval catchment (Carrière et al., 2020).** *Considering only the unvegetated parts of the catchments as contributing to the sediment yield,* *and the measured sediment density of 1700 kg/m$^3$, the average erosion rate is around 8 mm/yr (Mathys, 2006).*

18) Line 120: please clarify the measurement periods and years for each of these data sets. You say later on line 135 that soil temperature was recorded between 2005-2019, but the recording periods of sediment yield and rainfall are unclear. It would actually be useful if these study periods are mentioned in the abstract or introduction as well, earlier in the paper.

Reviewer #1 made similar comments; we therefore add a paragraph at the beginning of Section 3 to clarify the recording periods and we supply a supplementary table detailing measurement periods for soil temperature.

We used two different time periods for our analyses:

- 2003 to 2020 for the monthly sediment export analysis (4.1, 5.2: Figs. 3 - 5)
- 2005 to 2020 for the temperature indicator analysis (4.2, 5.3 : Figs. 6 - 9)

Because these two results are not directly correlated, we used the longer time range available for the sediment-export data to analyse the hysteresis cycle. However, when comparing annual sediment export with the temperature indicators (Fig. 6) we only used the data in the timespan 2005-2020 in order to be consistent with the time range of the soil temperature datasets.

19) Line 134: it is a bit unclear how many soil temperature sampling sites there were: was it four on each catchment (south-facing top, south-facing bottom, north-facing top, north-facing bottom?). In figure 1D, only 3 sites are marked as SF-top, SF-bot, and NF, so it's a bit confusing if for the NF, there were also top and bottom sites, and whether there was only one monitored hillslope per catchment, or one in each catchment (so 8 monitored sites in total?). Also, what is the elevation of this monitored hillslope with respect to the mean elevation of the catchment?

The paragraph starting line 134 contains the answers to these questions. In line 135 we mention that probes are implanted '*on opposite slopes in an inner meander of the Moulin Creek'*. Thus, there are 2 hillslopes in only one catchment. Additionally, line 136 clarifies the location of each probe site: '*probes located in bare black marls at uphill and downhill locations on north- and south-facing slopes'*. Confusion might come from Figure 1D; therefore we will add 'top' and 'bot' behind 'NF' in the revised version of the manuscript.
The mean elevation of the Moulin catchment is about 917m (Mathys, 2006) and the probes are located in the lower part of the catchment at an elevation around 867m.

Line 134: *Soil temperature has been recorded using several PT100 soil-temperature probes located on opposite slopes (867m elevation) in an inner meander of the Moulin Creek between August 2005 and December 2019 (i.e., four sites in total; Fig. 1D).*

20) Line 139-141: You say earlier that soil temperature was monitored between August 2005-December 2019. But then here you say "we specifically analysed soil temperatures during the winter season, from October 18[th] to 31[st]" – I guess this means that data was recording throughout the year but for this study you only focus on the winter temperatures? Or does this mean that only winter temperatures were recorded? Please clarify.

This is correct, as mentioned in line 139, we only used winter temperatures to study frost weathering because negative temperatures do not occur during spring, summer and early fall.

21) Line 148-153: show this relationship of air and soil temperature in the supplement – it's hard to follow a description of data that is not shown anywhere.

As suggested also by reviewer #1, we added figures and details in the supplementary material (Supplementary Figures 2 and 3) .

22) Line 152: what proportion of the winter season is snow-free?

In the Draix-Bléone catchments, snow fall occurs almost every year but these very small events accumulate less than 10 cm snow depth in general. Additionally, this thin layer rarely remains more than 1 day on south-facing slopes because of direct solar radiation, whereas it can stay a few days on North-facing slopes. Because these conditions are not very frequent, snow-cover periods have not been recorded. Only soil-temperatures records, on north-facing slopes, in the subsurface give approximate indications of snow-cover periods (e.g., Figure 2B).

Line 105: *The rainfall regime varies across seasons, with high-intensity rainfall events during spring/summer and lower-intensity but longer rainfall events in autumn.* ***Snow fall occurs almost every year but in small amounts (<10cm) and it melts quickly****. The mean annual temperature is 10.3 °C, with an annual variability between mean daily temperatures of approximatively 0.5 °C in winter and 20 °C in summer.*

23) Line 158: what previous studies? Please cite them.

Line 158: *We selected this range of thresholds based on inferences from previous studies* ***(Mathys, 2006)*** *and then used the value providing the best correlation to predict annual sediment-export values.*

24) Line 162: is that really your aim? The whole study seems focused to explore only one controlling factor, not test for other potential controlling factors.

This concluding sentence explains the general goal of the sediment-export anomaly calculation. To be consistent with the modifications made in the introduction, the following clarification has been added in the text:

Line 162: ***With the aim of quantifying the impact of frost weathering on sediment production, and of identifying the most relevant frost-weathering process, our objective is*** *to identify a controlling factor to explain this sediment-export anomaly*

25) Lines 210-211: these years of measuring periods of precipitation and sediment export should have been mentioned on the methods.

As we received comments from reviewer #1, we added a paragraph at the beginning of the Section 3.2.1 (Line 155).

Line 155: *First, we compared monthly sediment export and monthly rainfall to understand the seasonal dynamics of sediment transport in the Laval and Moulin catchments (Figs. 3 - 5). The analysis was performed for the period 2003 - 2019, during which sediment export was precisely recorded.*

26) Line 219: are there any general trends of increasing or decreasing precipitation or sediment export across the sampling years?

There is a large variability but no significant increasing or decreasing trend for precipitation or sediment export over the period we used (2003-2019).

27) Lines 342-343: why focusing only on small or marly catchments? It would be good to add some context and mention other studies that have shown that frost-cracking is an important control or modulator in sediment production (e.g. Hales and Roering's works on New Zealand, Delunel et al., 2010).

We are not sure that we fully understand this comment in the sense that Rengers et al (2020) studied a chalk hillslope rather than a marly catchment (Line 341). This part of the discussion also does not seem the right place to cite these other studies. However, we agree that they are relevant and therefore we will refer to them in the introduction as follows:

Line 76: *Frost-cracking has thus been identified as the major control on rock weathering in high Alpine environments (Hales and Roering, 2007; Delunel et al, 2010; Bennett et al., 2013; Draebing and Krautblatter, 2019).*

28) Lines 365-369: please include these figures in the supplementary info, it's hard to follow and blindly believe data that you can't see.

We will add in the supplementary material a figure and numerical results about the correlation between the FCI indicator and sediment export anomalies of the years n+1, n+2, n+3. We also modify the text as follows because some numerical results were slightly different than previously due to the initial choice to take the mean temperature value of each hillslope face.

Line 361: *These correlations between sediment production computed during a winter season and sediment yield for the directly following year (spring to autumn), together with the strongly varying dynamics of transport during the year discussed in the previous section, favour the hypothesis of rapid sediment export from the studied catchments, in contrast to the 3-year residence time of sediments in these catchments inferred by Jantzi et al. (2017). In order to test this hypothesis, we performed correlations between the frost-cracking intensity in a particular winter season (n) and the sediment-export anomaly of the first (n+1), second (n+2) and third (n+3) year after that season (i.e., in the n+3 case, if we consider the 2006-2007 winter season for the frost-cracking intensity, we compare it to the sediment-export anomaly for 2010) (Supplementary Figure 5). In all configurations, the correlation is weaker than the direct annual correlation that we observed ($R^2 = 0.76$) and the correlation weakens with increasing residence time (Suppl. Fig. 6); correlations for the years n+2 and n+3 are not significant . The ratio observed in sediment distribution during the spring/summer (74% suspended load / 26% bedload) and the rapid export of these fine sediments probably make the suspended load invisible in the estimation of sediment storage in the catchment, rendering the calculation of residence time complex.*

[Figure]

Supplementary Figure 6: *Regression analysis between frost-cracking intensity on the south-facing slope for the years (n =) 2007, 2008, 2009, 2010, 2014, 2015 and 2017 and the sediment-*

*export anomalies of A) the year n+1, B) the year n+2, C) the year n+3 (See text for explanation). Horizontal error bars refer to the difference between measurements at uphill and downhill locations (temperature measurements for 2017 were only available for the downhill location, thus uncertainty on frost-cracking intensity was computed as the average of the uncertainties from the others years); vertical error bars are ±2σ uncertainty in export anomaly. Red line shows ordinary least-squares (OLS) linear regression whereas* **blue** *line shows weighted linear regression following York et al. (2004). Weighted determination coefficients $R^2_w$ and associated normalized goodness-of-fit indicator $S_n = S / (n-2)$ are indicated for the weighted regression. Standard $R^2$ and associated p-value indicate significance of the ordinary least-squares (unweighted) regression.*

29) Line 413: the phrasing "later in the year"/ "an initial"/"followed by" is a bit confusing in the context of a hysteresis cycle. This would be clear if rephrasing to something like "from June to September", etc.

Line 409: *The annual hysteresis cycle (Fig. 4) shows an anti-clockwise pattern* **between June and August,** *and a clockwise pattern later in the year* **(September to December),** *suggesting a* **spring / early summer** *transport-limited regime followed by an autumn supply-limited regime in these catchments.*

FIGURES:

In general, all figures are on the lower-resolution side; higher-resolution, vectorized versions would be better for publication.

Figure 1: Some text is hard to read well because it's too pixelated, small, or a too fine font. In panel A, almost all text except the key and scale bar numbers are impossible to read. The cardinals are also impossible to read in all panels, I suggest just substituting for a bigger, bolder arrow pointing North. In panel B, all text is too small to be read easily, but the river names particularly. Panel C would be easier to read if the letters were bolder, or a white box is placed in the background and text switched to black. The black text in panel D is hard to read as well, it needs to be bolder, or have some outline in white, or a white box behind it.

We are surprised that the reviewer received low-resolution figures because we used the highest resolution possible for the insets of Figure 1. To improve the readability we increased the font, bolded some labels and changed the north arrow.

Figure 2: Does (A) have the same x-axis as B? If yes, please clarify in caption, if not, please add labels on axis.

We modify the caption as follows:

*Figure 1:* **Example of raw temperature series (1 measurement every 10 min).** *(A) Typical soil-temperature series recorded with four probes at different depths (from south-facing uphill location*; **time scale as in B**). *(B) Example of soil-temperature series (from south-facing downhill location) biased because of climatic conditions (snow cover), buried or loosened probes. A full year of temperature measurements is shown in Supplementary Figure 1.* **High**

*temperature values are observed at -1cm even in winter when black marl heats up during sunny periods.*

Figure 3: How have outliers been identified? Please explain on methods.

As we detailed in the caption to Fig. 3, the boxplot graphical representation indicates the mean, the median and the 1st and 3rd quartile (colour box limits). To define whiskers and outlier points we followed the Interquartile range (IQR) method of outlier detection, which commonly uses a scale of 1.5. This number controls the sensitivity of the whiskers range and is set to be close from what the Gaussian distribution with a 3σ range would give .

As these are standard methods we do not feel they need to be explicited in the methods section. However, we will add a short explanation to the figure caption:.

Line 578: *Black dots are outlier values (i.e., with values <Q1 - 1.5 IQR or > Q3 + 1.5 IQR).*

Figure 8: Choose a different colour for the weighted regression line that is not the same as the data points and error bars.

We will plot the data points in black and leave the weighted regression line in blue for Figures 7 and 8 in the revised version of the manuscript.